# TOUCH: Text-guided Controllable Generation of Free-Form Hand-Object Interactions

**Guangyi Han**[1,†], **Wei Zhai**[1,†], **Yuhang Yang**[1], **Yang Cao**[1], **Zheng-Jun Zha**[1,*]
[1] MoE Key Laboratory of Brain-inspired Intelligent Perception and Cognition,
University of Science and Technology of China
{hanguangyi@mail., wzhai056@, yyuhang@mail., forrest@, zhazj@}ustc.edu.cn

## Abstract

Hand-object interaction (HOI) is fundamental for humans to express intent. Existing HOI generation research is predominantly confined to fixed grasping patterns, where control is tied to physical priors such as force closure or generic intent instructions, even when expressed through elaborate language. Such an overly general conditioning imposes a strong inductive bias for stable grasps, thus failing to capture the rich diversity of daily HOI. To address these limitations, we introduce the new task of **Free-Form HOI generation**, which aims to generate controllable, diverse, and physically plausible HOI conditioned on fine-grained intent, including non-grasping actions like pushing, poking, and rotating. To support this task, we construct **WildO2**, the first large-scale, in-the-wild 3D HOI dataset, which includes non-grasping motions derived from internet videos; it contains 4.4k unique interactions across 92 intents and 610 object categories, each with detailed semantic annotations. Building on this rich dataset, we propose **TOUCH**, a three-stage framework centered on a multi-level diffusion model that facilitates fine-grained semantic control to generate versatile hand poses beyond grasping priors. This process leverages explicit contact modeling for conditioning and is subsequently refined with contact consistency and physical constraints to ensure realism. Comprehensive experiments demonstrate our method's ability to generate controllable, diverse, and physically plausible hand interactions representative of daily activities. Project page is https://guangyid.github.io/hoi123touch/.

## 1 Introduction

Hand-Object Interaction (HOI) is fundamental to expressing intent and executing tasks in human daily life, and the ability to generate controllable interactions is crucial for AR/VR, robotics, and embodied AI (Zheng et al., 2025). While existing HOI generation research has progressed from ensuring physical plausibility (Fang et al., 2020) to incorporating semantic controllability (Li et al., 2024b; Yang et al., 2023), its scope remains predominantly confined to a grasp-centric paradigm. The control signals in these methods, whether simple physical constraints like force closure or coarse high-level instructions (e.g., verb-noun pairs), are often overly general. This simplified conditioning imposes a strong inductive bias that primarily favors the generation of stable grasps (Taheri et al., 2020), sacrificing interaction diversity. Furthermore, even with more sophisticated control such as detailed natural language (via LLMs) (Zhang et al., 2025a;b), the underlying model designs and inherent inductive biases are still fundamentally geared towards generating only grasping interactions, driven by historical focus and prevailing representations. Consequently, these approaches lack the fine-grained control and inherent capability to capture the diverse non-grasping interactions found in the real world, including varied hand poses, contact details, and nuanced semantic intent.

To bridge the gap between the limited scope of current methods and the complexity of real-world interactions, we introduce the task of **Free-Form HOI Generation**. The goal is to break grasp-centric limitations and shift towards generating diverse interactions, including the vast array of non-grasping manipulations. This task emphasizes expressiveness and controllability in the generation

---

† Equal contribution. * Corresponding author.

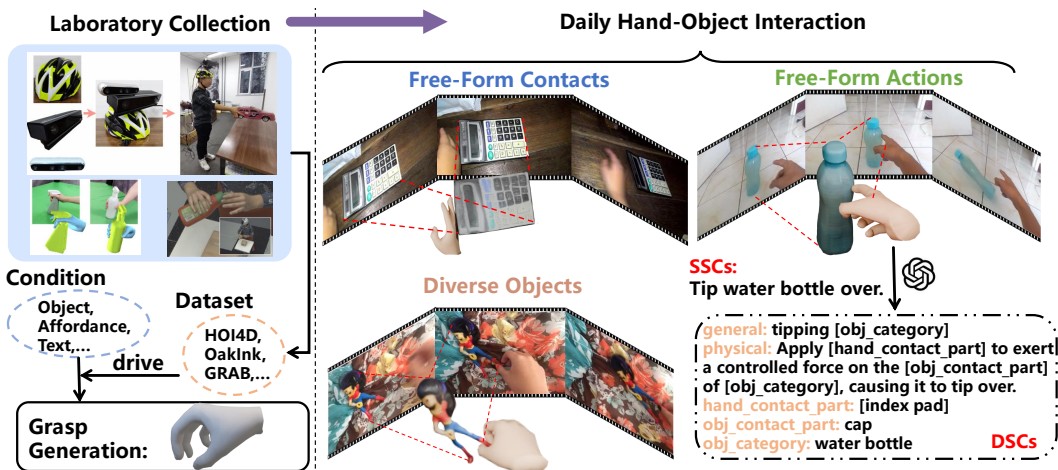

Figure 1: **Overview.** We extend HOI generation beyond laboratory "grasp" settings (left) toward broader daily HOI modalities (right), enabling the modeling of more human-like interactions. Our dataset WildO2, built from Internet videos, covers more contacts, more objects, and more actions, and is enriched with descriptive synthetic captions (DSCs) to support fine-grained semantic controllable HOI generation with our method, TOUCH.

process, aiming to synthesize interactions that are not only physically plausible but also semantically rich and truly adaptable to complex human intentions.

The core challenge of this task lies in two aspects: what to generate and how to generate it. The former pertains to spatial plausibility: the model must break free from restrictive grasping priors (e.g., palm position and orientation, contact region assumption (Ye et al., 2024; Jiang et al., 2021)) to explore a vast yet physically valid interaction space. To address this, we propose that contact relationships serve as a powerful cue to constrain this high-dimensional space, offering a more nuanced understanding of physically valid interactions. The latter pertains to semantic controllability: the model must accurately map fine-grained textual instructions to specific hand configurations and contact regions. The prior knowledge within Large Language Models (LLMs) offers a promising pathway for this guidance (Tang et al., 2023). A major obstacle to learning this complex mapping is the lack of 3D training data for diverse daily interactions, as existing datasets (Zhan et al., 2024; Fu et al., 2025) are mostly limited to lab-based grasping and object instances. Hardware and capture challenges make large-scale real-world 3D data collection difficult. In contrast, abundant 2D HOI videos online provide rich and realistic daily interaction behaviors.

To tackle the proposed task and challenges, we present TOUCH, a three-stage framework for controllable free-form HOI generation. First, we explicitly model the contact on the surfaces of the hand and the object separately by jointly encoding spatial point-cloud relations and semantic information, providing strong spatial priors to mitigate uncertainty from the high degrees of freedom in interaction position and pose. We further incorporate part-level hand modeling for more precise action control. Second, we employ a multi-level diffusion model with attention-based fusion of semantics and geometry: coarse-grained intent and global object geometry guide the early diffusion stages, while fine-grained text and local contact features refine detailed motions in deeper stages, enabling fine-grained semantic controllability. Finally, we introduce self-supervised contact consistency and physical plausibility constraints to optimize the generated interactions, ensuring realism and physical feasibility. Compared to prior methods restricted to grasp generation, TOUCH naturally generalizes to diverse free-form HOI such as pushing, pressing, and rotating.

Additionally, based on 3D object reconstruction (Xu et al., 2024), we introduce an automated pipeline to build the dataset WildO2 that jointly recovers and optimizes high-quality 3D hand-object interaction samples from internet videos annotated with interaction intent. By leveraging vision-language models (Bai et al., 2023b), we generate fine-grained semantic annotations, resulting in the 3D daily HOI dataset covering diverse interaction intents.

Our main contributions are: (1) We propose to extend the HOI from constrained grasping to a broader, more realistic, and more diverse set of daily interactions. (2) We propose TOUCH, a new

framework that can generate natural, physically reasonable, and diverse free-form HOI under fine-grained text guidance. (3) We build an automated pipeline and construct WildO2, an in-the-wild 3D dataset for daily HOI, providing a critical resource that enables future research in this domain. Extensive experiments demonstrate the superiority of TOUCH.

## 2 RELATED WORK

### 2.1 HAND-OBJECT INTERACTION DATASETS.

Existing 3D hand-object interaction (HOI) datasets are predominantly collected in controlled laboratory settings, relying either on physics-based simulation synthesis (Hasson et al., 2019) or motion capture systems to record real interactions (Hampali et al., 2020; Liu et al., 2022; Yang et al., 2022; Brahmbhatt et al., 2020). Although these datasets provide valuable support for modeling 3D HOI, they suffer from limited diversity due to constrained camera setups, a small number of participants, and a restricted set of object instances. In contrast, large-scale in-the-wild video datasets (Damen et al., 2020; Grauman et al., 2022; Shan et al., 2020) contain abundant HOI clips, but lack high-quality 3D annotations. Some studies have attempted to annotate subsets of these videos in 3D using object template-based optimization methods (Cao et al., 2021; Patel et al., 2022); however, due to the high diversity of open-set objects, scaling such approaches remains challenging.

### 2.2 TEMPLATE-FREE HOI RECONSTRUCTION.

The core bottleneck in reconstructing HOIs in the wild has long been the recovery of diverse object geometries. While existing template-free approaches (Fan et al., 2024; Ye et al., 2022) avoid predefined object model constraints, they are typically trained on limited datasets and exhibit poor generalization to novel objects. In recent years, multi-view diffusion models (Liu et al., 2023a) and large-scale reconstruction models (LRMs) (Hong et al., 2024) have enabled high-quality 3D mesh reconstruction directly from single images (Xu et al., 2024; Liu et al., 2024b) or text prompts (Poole et al., 2022), demonstrating strong generalization capabilities. Motivated by these advances, several HOI studies have explored image-to-3D reconstruction pipelines to handle open-set objects in the wild. However, due to severe hand occlusion, these methods often rely on image inpainting to complete occluded regions (Tian et al., 2025; Liu et al., 2024a; Wen et al., 2025; Liu et al., 2024c), or employ text-to-3D generation to align with coarse reconstruction results (Wu et al., 2024; Chen et al., 2025). Nonetheless, most of these pipelines depend on heuristic completion or registration strategies, resulting in limited geometric consistency with the input, and have yet to be validated at scale in an automated manner.

### 2.3 DATA-DRIVEN CONTROLLABLE HOI GENERATION.

In the evolution of HOI generation, interaction guidance has progressively advanced: from coarse control based on grasp type (Feix et al., 2015), to object-conditioned generation (Karunratanakul et al., 2020; Jiang et al., 2021), and further to task/action-level intent constraints (Christen et al., 2024; Yang et al., 2024b;a; Yu et al., 2025). To enhance physical plausibility, contact penetration loss and hand anatomical constraints have been widely adopted (Wei et al., 2024). Additionally, explicit modeling of hand part segmentation and contact relationships with objects has been shown to improve physical realism and detailed expression of interactions (Liu et al., 2023b; Zhang et al., 2024; Li et al., 2024a). Building on these efforts, we propose a multi-level controllable generation framework trained on our newly constructed daily HOI dataset, enabling finer-grained semantic intent control and the flexible generation of free-form HOIs that align with complex human intentions.

## 3 DATASET

### 3.1 DATA COLLECTION AND PROCESSING

Our goal is to construct a diverse dataset of 3D hand-object interactions from in-the-wild videos. A primary challenge in this process is the severe occlusion of the object by the hand, which compromises the quality of 3D object reconstruction. To address this, we introduce a semi-automated

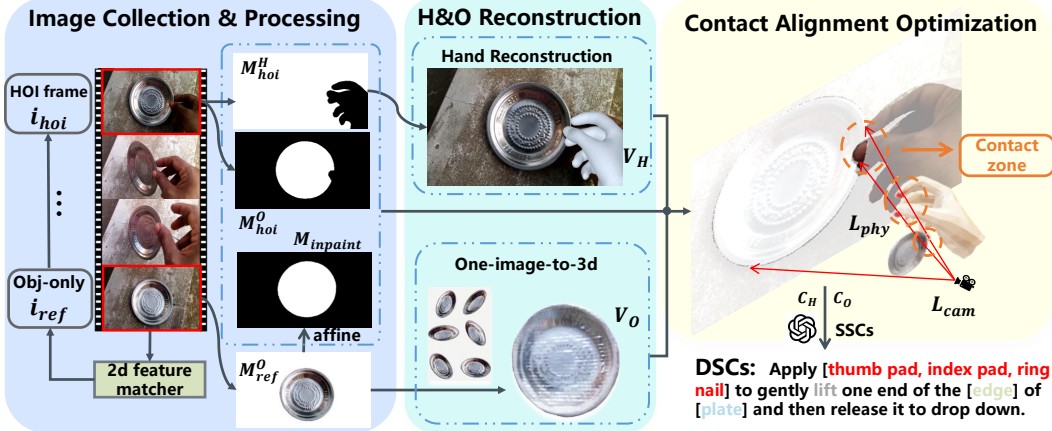

Figure 2: The proposed data reconstruction and annotation pipeline for WildO2. The process begins with O2HOI frame pair extraction from in-the-wild videos, followed by a three-stage pipeline for 3D reconstruction, camera alignment, and hand-object refinement that produces high-fidelity interaction data.

data reconstruction and annotation pipeline, centered around a novel Object-only to Hand-Object Interaction (O2HOI) frame pairing strategy.

We begin by filtering the Something-Something V2 dataset (Goyal et al., 2017), which is rich in goal-directed human actions, to obtain 8k single-hand, single-object interaction clips. For each clip, we automatically extract an O2HOI pair (details in Appendix): an object-only frame $I_{ref}$, where the object is unoccluded, and a corresponding interaction frame $I_{hoi}$. To obtain a complete object mask in the interaction frame, we segment the object in $I_{ref}$ using SAM2 (Ravi et al., 2024) and then transfer this mask to $I_{hoi}$ via a robust dense matching model (Edstedt et al., 2024), yielding $\mathbf{M}_{inpaint}$. This mask transfer strategy offers a distinct advantage over common alternatives: it avoids the geometric inconsistencies of diffusion-based inpainting (Liu et al., 2024a) while being significantly more scalable than manual completion (Wen et al., 2025). Consequently, our approach facilitates the automated, large-scale generation of high-fidelity 3D assets for reconstruction.

## 3.2 DATA RECONSTRUCTION PIPELINE

Based on the O2HOI pairs, we build a three-stage generation pipeline to recover 3D HOI.

**Stage 1: Initialization.** For each pair, we reconstruct a textured object mesh $\mathbf{V}^O_{recon}$ from the object-only frame $I_{ref}$ using an image-to-3D model (Xu et al., 2024). Concurrently, we estimate initial MANO (Romero et al., 2017) hand parameters $\mathbf{H}_{init}$ from the interaction frame $I_{hoi}$ using a state-of-the-art hand reconstruction method (Pavlakos et al., 2024).

**Stage 2: Camera Alignment.** A challenge arises from coordinate system misalignment: the object mesh $\mathbf{V}^O_{recon}$ is created in a canonical space of the object-only frame $I_{ref}$, while the hand exists in the camera space of the interaction frame $I_{hoi}$. To unify them, we align $\mathbf{V}^O_{recon}$ to an object-centric global coordinate system relative to the interaction frame by optimizing the camera projection matrix $\mathbf{K}$ and extrinsics $(\mathbf{R}, \mathbf{t})$. This is achieved by minimizing a camera alignment loss, $L_{cam}$, via differentiable rendering. The optimization proceeds in two phases: we initially use mask IoU, Sinkhorn (Cuturi, 2013) loss, and an edge penalty term (to prevent the object from moving out of view). Once the IoU surpasses a threshold, we introduce scale-invariant depth (Eigen et al., 2014) and RGB reconstruction losses for fine-tuning. The overall objective is formulated as:

$$\min_{\mathbf{K},\mathbf{R},\mathbf{t}} ; L_{cam} = L_{mask} + L_{sinkhorn} + L_{edge} + \lambda_{fine}(L_{depth} + L_{rgb}). \tag{1}$$

**Stage 3: Hand-Object Refinement.** With the aligned camera and object, we refine the initial hand parameters $\mathbf{H}_{init}$ to achieve physically plausible contact. Specifically, we cast rays from the camera center through pixels within the interaction mask $\mathbf{M}_{inpaint}$. The intersection points of these rays with the 3D hand and object geometries define a potential 3D contact zone. We then optimize $\mathbf{H}$ using a refinement objective $L_{align}$, which combines 2D evidence with 3D physical constraints: hand

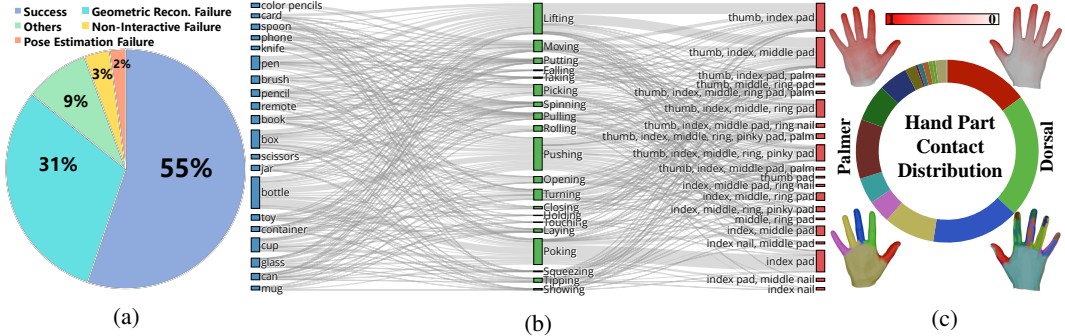

Figure 3: Dataset Statistics: (a) Breakdown of WildO2 reconstruction outcomes. (b) An illustration of the interplay between the most frequent object categories, interaction types, and hand contact regions. Object and action definitions are adapted and refined from (Goyal et al., 2017). Contact regions are derived based on our dataset analysis. (c) Specific segmentation of the 17 hand parts and their contact frequency distribution in the dataset, along with a contact heatmap of the entire hand.

mask IoU ($L_{\text{mask}}^{H}$), 2D joint reprojection error ($L_{\text{j2d}}$), an ICP loss on the 3D contact zone ($L_{\text{icp}}$), and physical constraints for contact, penetration, and anatomy based on (Yang et al., 2021).

$$\min_{\mathbf{H}}; L_{\text{align}} = L_{\text{mask}}^{H} + L_{\text{j2d}} + L_{\text{icp}} + L_{\text{phy}}, \quad L_{\text{phy}} = L_{\text{contact}} + L_{\text{pene}} + L_{\text{anatomy}} + L_{\text{self}}. \quad (2)$$

This pipeline yields 4,414 high-quality 3D hand-object interaction samples after a final stage of manual inspection and refinement, which constitute the ground truth of our dataset.

### 3.3 DATA ANNOTATION AND STATISTICS

We enrich our dataset with a multi-level annotation system, generating over 44k annotations. A statistical overview is provided in Fig. 3, with further details in the Appendix.

**3D Geometry and Transformation.** Each sample includes the final hand-object meshes ($\hat{\mathbf{V}}_H, \hat{\mathbf{V}}_O$) and the corresponding camera parameters derived from our generation pipeline. **Contact Maps.** We compute dense contact maps between the hand and object surfaces. To handle varying object scales, our method robustly identifies contact regions by combining relative and absolute distance thresholds with bidirectional nearest-neighbor filtering. **Multi-Level Language Descriptions.** We provide two levels of textual descriptions. We inherit the template-based Short Synthetic Captions (SSCs) from Something-Something V2 (e.g., "picking [Something] up"). Additionally, we use a Vision-Language Model (VLM) (Bai et al., 2023b) to generate more detailed Descriptive Synthetic Captions (DSCs), which are manually verified for quality and relevance. **Fine-Grained Hand Part Segmentation.** We segment the hand mesh into 17 parts, including finger pads, nails, knuckles, palmar, and the dorsal region. This partitioning scheme goes beyond the coarse divisions commonly used in grasp generation tasks (Hasson et al., 2019; Liu et al., 2023b)—which often focus only on contact on the inner hand—by also accounting for contact on the dorsal side. This fine-grained segmentation supports detailed local interaction analysis and facilitates alignment with the semantic descriptions in the DSCs.

## 4 METHOD

This work aims to generate natural and physically plausible hand-object interaction (HOI) poses, parameterized by $\mathbf{H}$, along with corresponding contact maps $\mathbf{C}_H$ and $\mathbf{C}_O$, conditioned on a multi-level textual prompt $\mathbf{T}$ and an object mesh $\mathbf{V}_O$. To tackle this problem, we propose a three-stage framework, as illustrated in Fig. 4. Specifically, the Contact Map Prediction module (Sec. 4.1) infers the potential contact regions on the hand and object surfaces based on the text and object geometry. The Multi-Level Conditioned Diffusion module (Sec. 4.2) synthesizes a coarse hand pose by integrating coarse-to-fine textual and geometric features within a diffusion framework, ensuring alignment with multi-level constraints. Finally, the Physical Constraints Refinement module (Sec. 4.3) further optimizes the coarse pose to enhance contact realism and prevent penetrations.

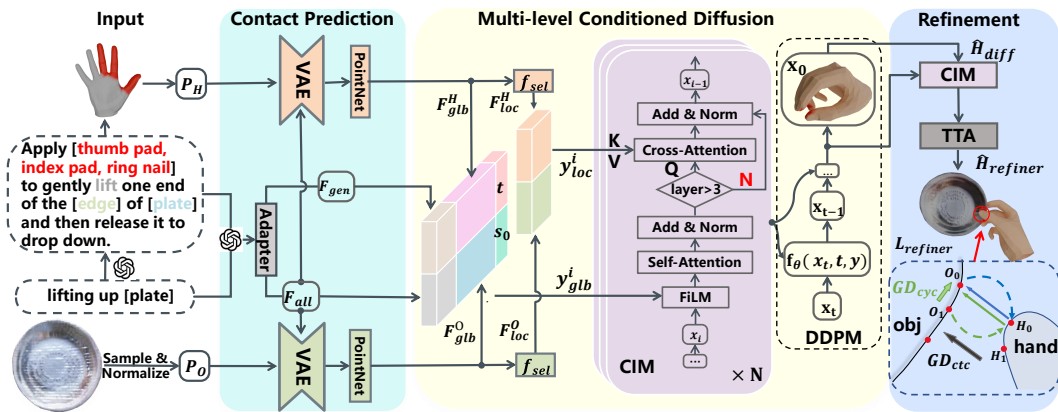

Figure 4: Overview of our three-stage framework TOUCH for generating hand-object interactions from multi-level text prompts and object meshes. CIM stands for the Condition Injection Module.

## 4.1 CONTACT MAP PREDICTION

To generate diverse interactions beyond simple grasping, we design two independent yet similar CVAEs (Sohn et al., 2015) to generate binary contact maps for the object and the hand, respectively. For the object branch, we sample a point cloud $\mathbf{P}_O \in \mathbb{R}^{N_O \times 3}$ ($N_O = 3000$) from its mesh $\mathbf{V}_O$, normalize it, and record the scale factor $s_O$. We use PointNet (Qi et al., 2016) to extract its geometric features, which are concatenated with $s_O$ to form the object condition $\mathbf{F}_O$. For the hand branch, we generate a canonical point cloud $\mathbf{P}_H^0 \in \mathbb{R}^{N_H \times 3}$ ($N_H = 778$) from MANO's zero pose and shape parameters $\mathbf{H}^0$. This point cloud, combined with a hand-part mask initialized from the fine-grained text $\mathbf{T}_{DSC}$, is processed by PointNet to obtain the hand condition $\mathbf{F}_H$. This design integrates the topological structure of the point clouds with text-guided emphasis on interaction-relevant hand regions. Both CVAEs are trained conditioned on their respective geometric features ($\mathbf{F}_O, \mathbf{F}_H$) and a shared text feature $\mathbf{F}_{DSC} = f_{text}(\mathbf{T}_{DSC})$, which is extracted using the Qwen-7B (Bai et al., 2023a) processed through a lightweight adapter. The optimization objective is a composite loss function:

$$L_{contact} = L_{focal} + L_{dice} + \beta L_{KL}, \tag{3}$$

where $L_{focal}$ and $L_{dice}$ supervise the contact prediction, and $L_{KL}$ structures the latent space. During inference, under the conditional features ($\mathbf{F}_O, \mathbf{F}_H, \mathbf{F}_{DSC}$), the model samples from a Gaussian prior $z \sim \mathcal{N}(0, I)$ and decodes it to produce the predicted binary contact maps $\hat{\mathbf{C}}_O \in \{0,1\}^{N_O \times 1}$ and $\hat{\mathbf{C}}_H \in \{0,1\}^{N_H \times 1}$.

## 4.2 MULTI-LEVEL CONDITIONED DIFFUSION

The core of our method is a Transformer-based Denoising Diffusion Probabilistic Model (DDPM) (Ho et al., 2020) that synthesizes hand pose parameters $\hat{\mathbf{H}}$ conditioned on the object point cloud $\mathbf{P}_O$, multi-level text $\mathbf{T}$, and predicted contact maps $\hat{\mathbf{C}}$. Instead of predicting noise, our model $f_\theta$ is trained to directly predict the denoised data $\hat{\mathbf{x}}_0 = f_\theta(\mathbf{x}_t, t, \mathbf{y})$, optimized with an L2 loss on the pose parameters: $\mathcal{L}_{diff} = \mathbb{E}_{t,\epsilon}\left[\|\hat{\mathbf{x}}_0 - \mathbf{x}_0\|^2\right]$.

**Condition Generation: Transformer Inputs.** To achieve precise control, our model extracts multi-level conditional features from both geometric and textual modalities. On the geometric side, we use PointNet to extract global features $\mathbf{F}_{glb}^O, \mathbf{F}_{glb}^H$ and point-wise local features from the object point cloud $\mathbf{P}_O$, the initial hand point cloud $\mathbf{P}_H^0$, and the predicted contact maps $\hat{\mathbf{C}}$ from the previous stage. To focus on interaction regions, we leverage $\hat{\mathbf{C}}$ to adaptively select features of $N_{loc}^O = 128$ object points and $N_{loc}^H = 64$ hand points near contact areas, yielding $\tilde{\mathbf{F}}_{loc}^O$ and $\tilde{\mathbf{F}}_{loc}^H$. On the textual side, we utilize $f_{text}$ to extract both coarse-grained $\mathbf{F}_{qwen}^{SSC} = f_{text}(\mathbf{T}_{SSC})$ and fine-grained $\mathbf{F}_{qwen}^{DSC}$ text features.

**Conditional Injection: Coarse-to-Fine Control.** We inject these features into the $N_{inj} = 8$ blocks of our Transformer model in a hierarchical, coarse-to-fine fashion. This design ensures that global

context, defined by SSCs and global geometry, shapes the overall pose in early denoising stages, while local details, defined by DSCs and contact-point features, are refined in later stages. Specifically, for the $i$-th Transformer block:

Early Stages ($i < 4$): Global context is injected, with no local features.

$$\mathbf{y}^i_{\text{glb}} = \text{concat}(\mathbf{F}^O_{\text{glb}}, \mathbf{F}^H_{\text{glb}}, s_O, \mathbf{F}^{SSC}_{\text{qwen}}, t), \quad \mathbf{y}^i_{\text{loc}} = \emptyset. \tag{4}$$

Later Stages ($4 \leq i < N_{\text{inj}}$): Local details are injected, switching to fine-grained conditions.

$$\mathbf{y}^i_{\text{glb}} = \text{concat}(\mathbf{F}^H_{\text{glb}}, s_O, \mathbf{F}^{DSC}_{\text{qwen}}, t), \quad \mathbf{y}^i_{\text{loc}} = \text{concat}(\tilde{\mathbf{F}}^O_{\text{loc}}, \tilde{\mathbf{F}}^H_{\text{loc}}). \tag{5}$$

To prevent over-reliance on any single condition and enhance robustness, we randomly drop each component of the global condition with a 10% probability during training. Within each block, the global condition $\mathbf{y}^i_{\text{glb}}$ modulates the main features via FiLM (Perez et al., 2018), while the local condition $\mathbf{y}^i_{\text{loc}}$ is integrated through cross-attention to provide fine-grained spatial cues. This dual mechanism effectively decouples global contextual guidance from local geometric refinement. Finally, the updated latent goes through self-attention and a Feed-Forward Network (FFN).

**Training Loss.** To improve training stability and spatial alignment, we introduce two auxiliary losses alongside the primary diffusion loss $\mathcal{L}_{\text{diff}}$. A global pose loss directly supervises the hand's global rotation $\mathbf{r}_{\text{rot}}$ and translation $\mathbf{T}$ to prevent overall pose drift, an issue exacerbated when directly regressing $\mathbf{H}$, which comprises parameters with disparate numerical ranges (e.g., shape $\boldsymbol{\beta}$, pose $\boldsymbol{\Theta}$, $\mathbf{r}_{\text{rot}}$, $\mathbf{T}$). A distance map loss ensures precise contact by supervising the distance map $\mathbf{d}_{\text{map}} \in \mathbb{R}^{21 \times N_O}$ from the 21 hand joints to the object surface. The final objective is a weighted sum:

$$\mathcal{L}_{\text{total}} = \mathcal{L}_{\text{diff}} + \lambda_{\text{global}}\left(|\hat{\mathbf{r}}_{\text{rot}} - \mathbf{r}^{\text{gt}}_{\text{rot}}| + |\hat{\mathbf{T}} - \mathbf{T}^{\text{gt}}|\right) + \lambda_{\text{dmap}}|\hat{\mathbf{d}}_{\text{map}} - \mathbf{d}^{\text{gt}}_{\text{map}}|. \tag{6}$$

## 4.3 Physical Constraints Refinement

To address the common issue of global pose drift in free-form HOI generation, where the hand often fails to make contact with the object, we introduce an efficient physical refinement module. This module is powered by a refiner network, $f_{\text{refiner}}$, which inherits the Transformer architecture of our diffusion model. The process begins with a single forward pass to rapidly correct the global positioning of the initial pose $\hat{\mathbf{H}}_{\text{diff}}$, establishing primary physical contact. Subsequently, this corrected pose undergoes $N_{\text{tta}}$ iterations of test-time optimization (TTA) to fine-tune local contact details, such as finger placements.

The entire optimization is guided by our self-supervised cycle-consistency loss ($\mathcal{L}_{cyc}$), which enforces bidirectional mapping consistency between hand and object contact surfaces. The core idea is that a hand contact point, after being mapped to the nearest object point via $\Phi$ (hand-to-object), should map back to its original location via the reverse mapping $\Psi$ (object-to-hand), and vice versa. This loss acts as a powerful regularizer, effectively reducing the ambiguity inherent in the mappings. We combine this with $\mathcal{L}_{\text{phy}}$ (see Eq. 2). The total refinement loss is defined as:

$$\mathcal{L}_{\text{refiner}} = \mathcal{L}_{\text{phy}} + \lambda_{cyc}(\mathbb{E}_{\mathbf{P}_h \in \mathbf{P}_{C_H}}||\Psi(\Phi(\mathbf{P}_h)) - \mathbf{P}_h||_1 + \mathbb{E}_{\mathbf{P}_o \in \mathbf{P}_{C_O}}||\Phi(\Psi(\mathbf{P}_o)) - \mathbf{P}_o||_1). \tag{7}$$

## 5 Experiments

### 5.1 Experimental Settings

Our experiments are conducted on the WildO2 dataset. For each hand part contact category, we perform a random 4:1 split, yielding approximately 3.7k training and 677 test samples. To address the long-tailed distribution of hand part labels, we aggregate 10 less frequent hand part categories and then apply resampling using unique 7-bit labels to balance the data. The model is trained for 1000 epochs using the Adam optimizer with a learning rate of 1e-4 and a batch size of 128. The diffusion model's parameters are frozen during the training of the refiner module. We evaluate our method from four perspectives: (1) Contact Accuracy, assessed by IoU and F1-score against ground-truth contacts parts. (2) Physical Plausibility, measured by Mean Per-Vertex Position Error (MPVPE), Penetration Depth (PD), and Penetration Volume (PV). Note that unlike works focusing on grasping (Jiang et al., 2021), we do not employ physics engine-based stability simulation metrics, as our scope of interactions is broader than force-closure grasps. (3) Diversity, quantified by entropy and cluster size. (4) Semantic Consistency, evaluated using a point cloud-based FID (P-FID) (Nichol et al., 2022), VLM assisted evaluation, and a perceptual score (PS) from 10 users.

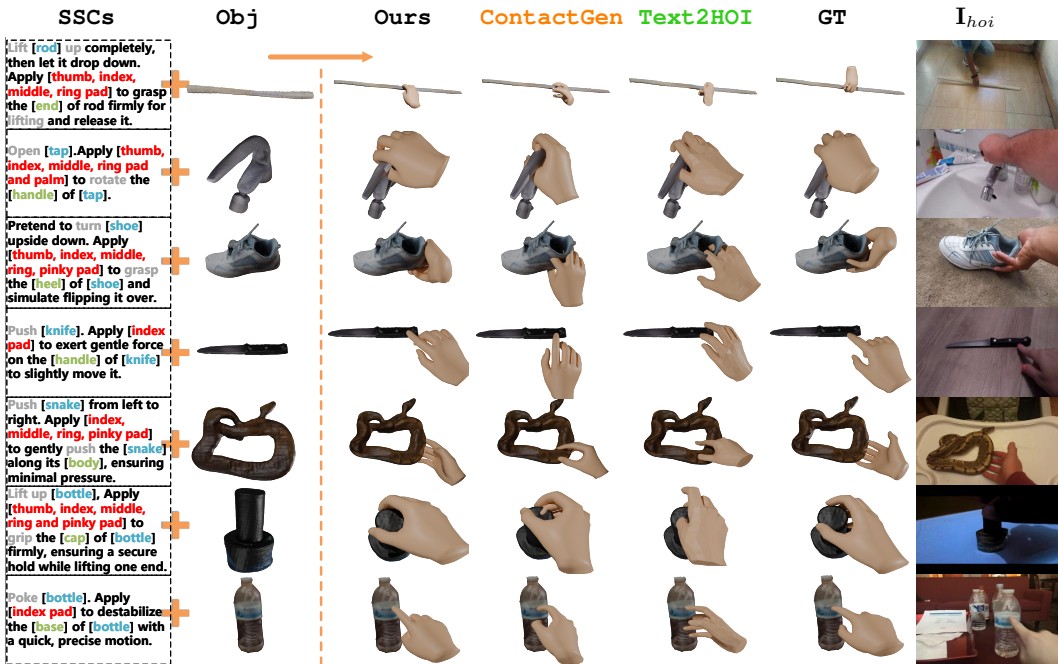

Figure 5: **Visualization Results.** Comparisons of different methods on samples from the WildO2 test set. Each sample consists of SSCs and an object mesh as input, with the output being an interactive hand pose. The last row shows the original authentic 2D HOI frame from internet videos.

## 5.2 COMPARISONS

As existing methods have not explored fine-grained controlled HOI generation, we select two representative types of baselines: (1) *ContactGen* (Liu et al., 2023b): an object-conditioned multi-layer CVAE using coarse hand part labels. (2) *Text2HOI* (Cha et al., 2024): a transformer-based conditional diffusion model guided by coarse text conditions. We remove its temporal axis and adapt it for our setting. Compared to typical grasping datasets, hand poses in WildO2 exhibit higher degrees of freedom. Both baseline methods exhibit noticeable overall hand drift. To ensure fair comparison, we also augment them with an optimization-based post-processing module to correct hand poses. Experimental results in Tab. 1 show that our method outperforms baselines across most metrics. Visual results in Fig. 5 further demonstrate that our method generates more realistic HOI poses that better align with input text descriptions.

| Method | Contact Acc. | | Physical Plausibility | | | Diversity | | Semantic Consistency | | |
|---|---|---|---|---|---|---|---|---|---|---|
| | P-IoU↑ | P-F1↑ | MPVPE↓ | PD↓ | PV↓ | Ent.↑ | CS↑ | P-FID↓ | VLM↑ | PS↑ |
| ContactGen | 0.620 | 0.730 | 5.46 | 1.296 | 7.37 | 2.85 | 4.93 | 6.08 | 4.8 | 6.3 |
| Text2HOI | 0.711 | 0.795 | 4.69 | 1.239 | 4.93 | 2.85 | 5.20 | 15.72 | 6.5 | 7.5 |
| **Ours** | 0.776 | 0.844 | 2.97 | 0.932 | 2.67 | 2.93 | 5.40 | 4.13 | 7.1 | 8.8 |

Table 1: **Quantitative comparison on HOI synthesis.** We evaluate all methods on our test set using comprehensive metrics covering physical plausibility, contact accuracy, diversity, and semantic consistency. ↑ indicates higher is better, ↓ indicates lower is better.

## 5.3 ABLATION STUDY

To evaluate the effectiveness of our contact-guided generation of spatial relations in HOI and the coarse-to-fine text control design, we conduct ablation studies as shown in Tab. 2, ✗ means without. For clarity, TTA is disabled. We argue for the primacy of contact metrics, as penetration metrics, PD and PV are meaningful only after hand-object contact is established; otherwise, they can be misleading. This is starkly exemplified by the "✗ refiner" variant, which scores poorly on contact yet achieves deceptively low PD/PV values simply because the generated hand drifts away from the

| Metrics | P-IoU↑ | P-F1↑ | MPVPE↓ | PD↓ | PV↓ | P-FID↓ |
|---|---|---|---|---|---|---|
| **Ours(✗ TTA)** | 0.728 | 0.805 | 3.00 | 1.093 | 4.82 | 4.84 |
| ✗ hoc. | 0.492 | 0.611 | 4.93 | 1.330 | 5.50 | 5.41 |
| ✗ refiner | 0.513 | 0.621 | 5.05 | 0.723 | 2.98 | 5.84 |
| ✗ $L_{cyc}$. | 0.702 | 0.787 | 3.00 | 1.100 | 5.29 | 5.79 |
| ✗ mul. | 0.525 | 0.631 | 5.00 | 1.464 | 6.52 | 6.84 |
| ✗ $T_{DSC}$ | 0.698 | 0.784 | 3.02 | 1.119 | 5.28 | 6.09 |
| ✗ $T_{SSC}$ | 0.687 | 0.778 | 2.92 | 1.119 | 5.17 | 5.52 |
| CLIP | 0.713 | 0.798 | 2.87 | 1.136 | 4.85 | 4.84 |
| BERT | 0.705 | 0.790 | 2.91 | 1.182 | 4.99 | 6.08 |
| MPNet | 0.704 | 0.788 | 2.87 | 1.114 | 5.06 | 6.02 |

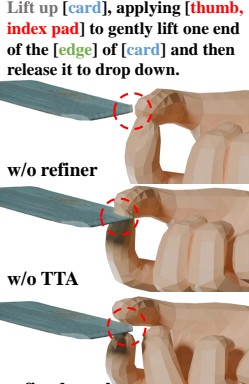

Table 2: Analyzes the contributions of various components, including the absence of $M_O$ and $M_H$ (hoc.) for guiding spatial relationship generation, the multi-level network structure (mul.), and the multi-level text (✗ $T_{DSC}$, ✗ $T_{SSC}$).

Figure 6: Contact guidance visualization.

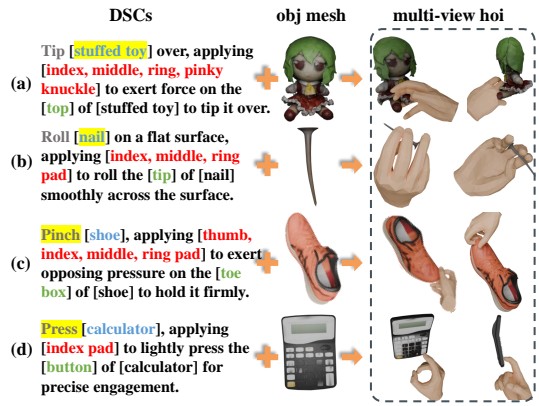

Figure 7: **Out-of-Domain Generalization on Objaverse.** (a,b) Out-of-domain CAD models; (c,d) Verbs outside our primary annotated intents.

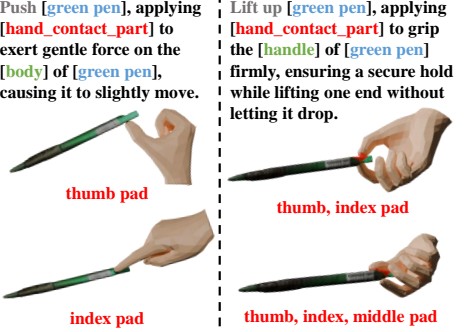

Figure 8: **Controllable Semantic Generation.** Results for a green pen, demonstrating diverse and plausible hand poses generated by specifying different control signals, including interaction intent and contact geometry.

object, thus avoiding interaction entirely. This distinguishes our complex task from traditional grasping, where contact is facilitated by priors. The consistent degradation in contact performance upon removing any module confirms their synergistic importance. Fig. 6 offers a qualitative visualization of the step-by-step improvements afforded by our contact guidance.

## 5.4 DISCUSSION

### 5.4.1 ABLATION ON TEXT ENCODERS.

We replace the text encoder with other common token- or sentence-level encoders (e.g., CLIP (Radford et al., 2021), BERT (Devlin et al., 2019), MPNet (Song et al., 2020)) to analyze their impact on generation quality. Results indicate that Qwen-7B offers better performance in capturing fine-grained semantic details, detailed in Tab. 2.

### 5.4.2 OUT-OF-DOMAIN GENERATION.

To evaluate generalization, we sample novel object CAD meshes from the large-scale 3D dataset Objaverse (Deitke et al., 2023), utilizing LLMs to generate captions that emulate our DSC format. As visualized in Fig. 7, our approach successfully produces plausible interaction poses for these out-of-domain models, demonstrating strong generalization capability.

### 5.4.3 SEMANTIC CONTROLLABILITY

**Controllable Semantic Generation.** Our model demonstrates strong controllability and high semantic faithfulness in interpreting fine-grained user intents. As shown in Fig. 8, by varying textual control signals (such as contact regions and interaction semantics (e.g., Push/Lift) for an object), the model successfully produces diverse and physically plausible hand poses, validating its ability to execute multi-faceted semantic directives.

**Semantic Nuances of Force Expression.** Although the model does not explicitly model physical forces, it learns to associate force-related terms like "firmly" and "gently" with contact geometry. It generates larger, denser contacts for "firm" prompts and more marginal, sparser contacts for "gentle" ones, as shown in Fig. 9. Quantitative analysis on WildO2 confirms this finding, revealing a 22-25% larger average contact area for "firm/tight" interactions.

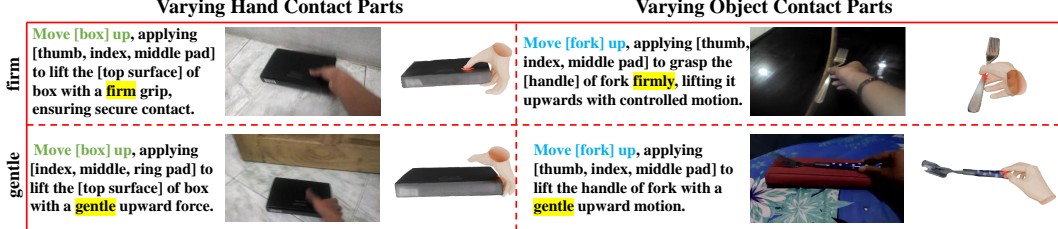

Figure 9: **Interpreting force-related semantics.** The model associates "firm" prompts (Top Row) with larger contact areas and more encompassing interactions, while mapping "gentle" prompts (Bottom Row) to sparser, more marginal contact points.

## 6 CONCLUSION

In this paper, we addressed the limitations of grasp-centric approaches by introducing the Freeform HOI Generation task. Our work expands the synthesis paradigm beyond simple grasping to a broader, more semantically expressive spectrum of interactions. To support this, we built an automated pipeline to construct WildO2, an in-the-wild 3D dataset for daily HOIs, providing a critical resource to enable future research in this domain.

**Limitations and Future Directions.** Our framework currently focuses on static HOI snapshots, which inherently limits its ability to capture the temporal dynamics of an interaction process. While our pipeline offers rapid expansion, the current dataset scale also presents an area for future growth. In the future, we plan to extend our work to dynamic sequences by leveraging large-scale video datasets and incorporating 6-DoF object pose estimation, thus modeling the entire human-environment interaction process.

## 7 ACKNOWLEDGMENTS

This work is supported by the National Natural Science Foundation of China (NSFC) under Grants 62225207, 62436008, 62306295, 62576328, and 625B2175. The AI-driven experiments, simulations and model training were performed on the robotic AI-Scientist platform of Chinese Academy of Sciences.

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

# A APPENDIX

## A.1 MORE EXPERIMENT RESULTS

### A.1.1 MORE ABLATION EXPERIMENTS

**Action-Specific Analysis** We performed a per-category statistical analysis of the metrics for the most frequent verb categories in the test set, as detailed in Tab. 3. The results reveal significant variations across different verb categories, reflecting the distinct characteristics of each action.

|  | Push | Poke | Lift | Pick | Move | Turn | Pull | Put | Open | Roll | Spin |
|---|---|---|---|---|---|---|---|---|---|---|---|
| P-IoU | 0.762 | 0.811 | 0.805 | 0.752 | 0.773 | 0.725 | 0.830 | 0.753 | 0.778 | 0.739 | 0.763 |
| P-F1 | 0.833 | 0.852 | 0.872 | 0.834 | 0.844 | 0.821 | 0.891 | 0.828 | 0.866 | 0.819 | 0.834 |
| MPVPE | 3.098 | 3.458 | 2.724 | 2.511 | 2.618 | 3.014 | 2.301 | 3.034 | 2.621 | 2.653 | 3.844 |
| PD | 1.135 | 0.391 | 0.568 | 1.127 | 0.811 | 1.462 | 1.214 | 1.105 | 1.206 | 1.294 | 0.892 |
| PV | 3.025 | 0.922 | 1.716 | 2.814 | 2.842 | 5.718 | 3.440 | 3.656 | 4.081 | 1.652 | 3.052 |

Table 3: Per-category performance metrics for the most frequent verb categories in the test set.

For instance, the Poke category, which typically involves a singular or highly localized contact area, consequently yields higher scores on contact-related metrics such as P-IoU and P-F1. However, this action imposes fewer constraints on the overall hand orientation and the configuration of non-contacting fingers, leading to a higher MPVPE for the generated hand poses. Conversely, the Lift category, which closely resembles a simple grasping motion, is generally less complex. As a result, it demonstrates strong performance across most metrics, which corroborates our earlier assertion regarding the relative simplicity of grasping operations. In general, for a given verb, greater diversity in the associated actions correlates with increased learning difficulty for the model.

**Layer Injection Strategy** We conducted an ablation study on the coarse-to-fine layer injection strategy (Tab. 4) to justify the chosen 4/4 split against alternatives (e.g., 0/8, 2/6, 6/2).

| Split Config | P-IOU ↑ | P-F1 ↑ | MPVPE ↓ | PD ↓ | PV ↓ |
|---|---|---|---|---|---|
| 0/8 (✗ mul) | 0.766 | 0.836 | 2.97 | 0.950 | 2.55 |
| 2/6 | 0.767 | 0.839 | 2.97 | 0.942 | **2.52** |
| **4/4 (Ours)** | **0.776** | **0.844** | **2.89** | **0.932** | 2.67 |
| 6/2 | 0.650 | 0.747 | 3.64 | 1.026 | 4.46 |

Table 4: **Ablation on Coarse/Fine Layer Split.** Comparison of different layer allocation strategies for the coarse-to-fine injection. The 4/4 split yields the best trade-off between semantic structure and geometric precision.

The results demonstrate two key points: (1) Optimal Balance: The 4/4 configuration achieves the best overall performance, confirming that a balanced injection strategy effectively handles both global semantics and local contact details. (2) Fine-grained Priority: The 2/6 split (late fine-grained injection bias) significantly outperforms the 6/2 split (early coarse injection bias). This indicates that for generating free-form HOI, conditioning on fine-grained information is more critical than early global coarse cues.

**Impact of Detailed Contact Directives.** To qualitatively illustrate the findings from our ablation study (Tab. 2), we visualize the model's behavior when fine-grained contact directives are omitted. As shown in Fig. 11, without this guidance, the model tends to collapse to common grasping

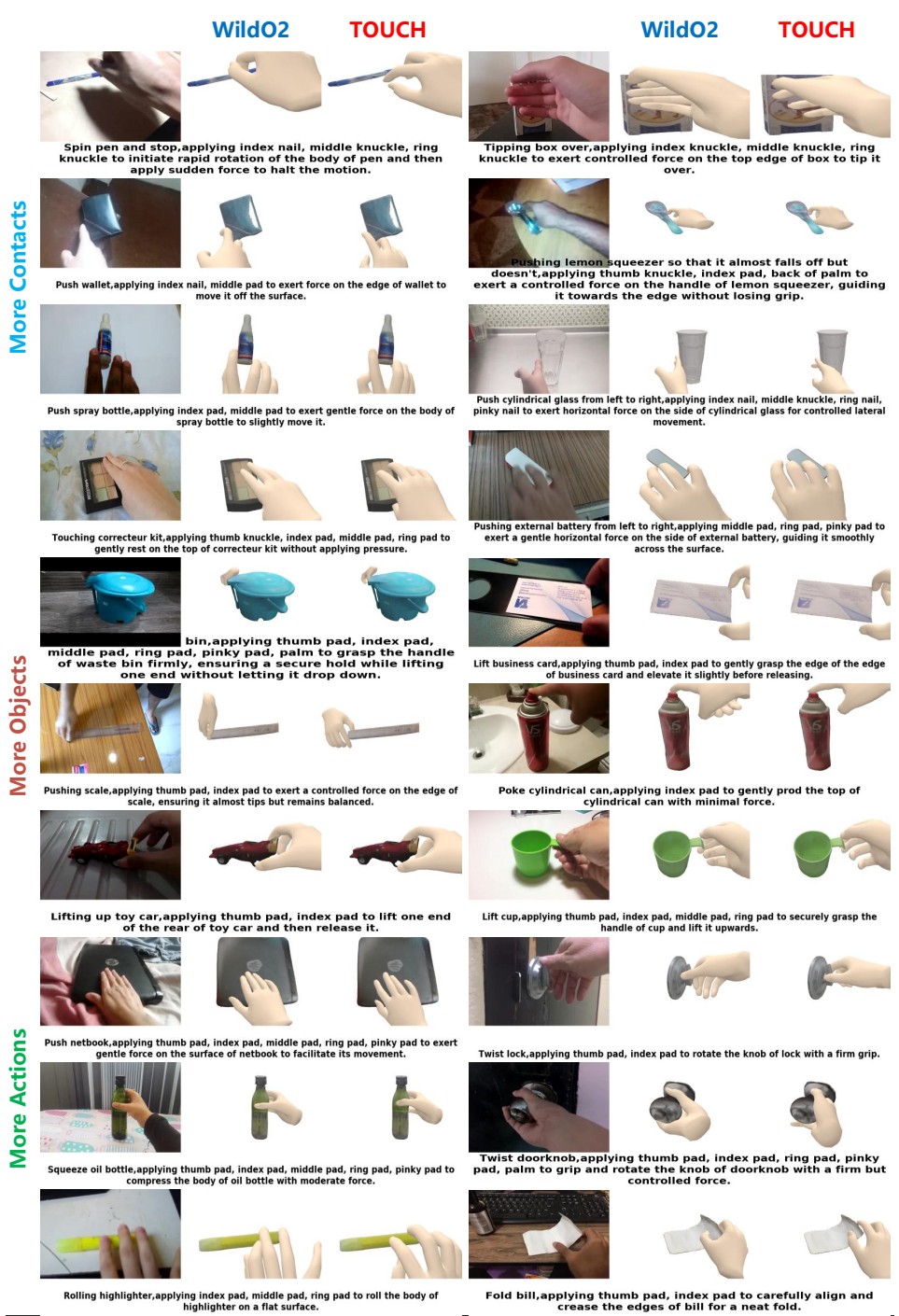

Figure 10: **Visualization Results.** More samples from the WildO2 test set. For each sample, the leftmost image represents the hand-object interaction frame ($I_{hoi}$) extracted from Internet videos. The middle image shows our reconstruction of $I_{hoi}$, which serves as the ground truth for WildO2. The rightmost image illustrates the free-form HOI generated by our method, TOUCH, conditioned on object meshes and DSCs. The bottom row contains the corresponding DSCs.

patterns, even for non-grasping prompts like "put tea bag on a surface". This underscores the necessity of detailed contact information for generating nuanced and accurate interactions beyond simple grasps.

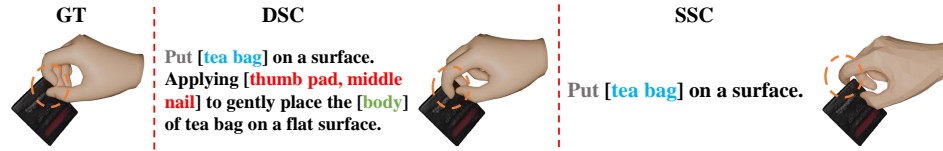

Figure 11: **Comparison of generations with and without detailed contact guidance. (Left)** Given a Short Synthetic Caption (SSC) that omits contact information, the model defaults to a generic and incorrect grasping pose. **(Right)** With a Detailed Synthetic Caption (DSC) specifying the intended contact points, the model successfully generates the correct, nuanced "placing" interaction.

### A.1.2 MORE COMPARISON EXPERIMENTS

We evaluate our method on the OakInk dataset, as detailed in Tab.5. Using its CapGrasp (Li et al., 2024b) annotations and computed part contact information, we construct fine-grained text conditions through summarization via Qwen (Bai et al., 2023a) and verification.

| Method | Contact Acc. | | Physical Plausibility | | | Diversity | | Semantic |
|---|---|---|---|---|---|---|---|---|
| | P-IoU↑ | P-F1↑ | MPVPE↓ | PD↓ | PV↓ | Ent.↑ | CS↑ | P-FID↓ |
| ContactGen | 0.654 | 0.769 | 6.69 | 0.790 | 16.59 | 2.93 | 5.33 | 11.25 |
| Text2HOI | 0.778 | 0.860 | 5.49 | 0.807 | 7.05 | 2.84 | 3.48 | 4.95 |
| **Ours** | 0.812 | 0.882 | 4.49 | 0.939 | 5.89 | 2.92 | 3.50 | 3.21 |

Table 5: **Quantitative comparison on hand-object interaction synthesis in OakInk-Shape dataset.** ↑ indicates higher is better, ↓ indicates lower is better.

### A.1.3 FAILURE CASES ANALYSIS

Despite its overall strong performance, our analysis reveals several recurring failure modes in the TOUCH model. These can be categorized into four main types, as illustrated in Fig. 12:

- **Pose Bias:** The model often defaults to a grasping pose, even when textual cues contradict this action (e.g., prompting a gentle touch without a thumb).
- **Orientation Error:** The generated hand exhibits an incorrect or unnatural palm orientation relative to the object.
- **Contact Mismatch:** The guided contact between the hand and the object is misplaced, leading to physically implausible interactions.
- **Penetration Artifacts:** The model produces significant penetration of the fingertips into the object's surface.

### A.2 DETAILS OF WILDO2 ACQUISITION

### A.2.1 SCALABILITY OF THE PIPELINE

Our pipeline's design emphasizes three key properties: **evolvability**, allowing it to improve with stronger upstream models; **scalability**, enabling rapid data reconstruction from web-scale clips approximately less than 2 minutes per case; and **controllability**, ensured by our rigorous human-in-the-loop quality control.

**Evolvability with Upstream Image-to-3D Models.** Our pipeline's performance is directly coupled with the upstream image-to-3D model, a modular design that allows us to benefit from ongoing advancements. We demonstrate this by replacing our default backbone, InstantMesh, with the more powerful Hunyuan3D 3.0 (Team, 2025) As shown in Fig. 13, this upgrade yields significant improvements. For samples that were already successfully reconstructed (first two rows), the new backbone enhances geometric details, such as refining object edges and correctly modeling the internal cavity

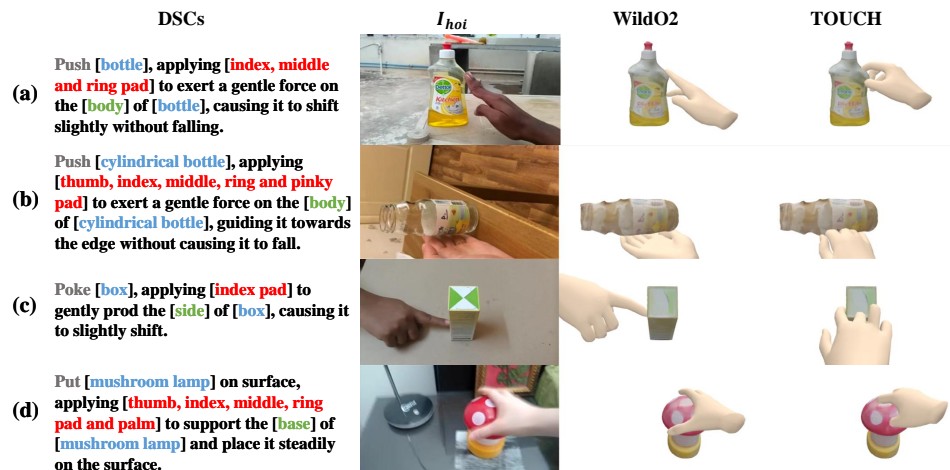

Figure 12: **Qualitative examples of typical failure modes of the TOUCH model.** Each panel corresponds to a category discussed in the text: (a) a bias towards grasping poses, (b) incorrect palm orientation, (c) misplaced contact guidance, and (d) significant fingertip penetration.

of a cup. More importantly, it successfully reconstructs instances that had previously failed (third and fourth rows), thereby increasing the overall success rate of our pipeline. Nevertheless, challenges persist even for advanced models, such as reconstructing transparent objects or fine-grained structures, highlighting the continued potential for future advancements to further enhance the quality of WildO2.

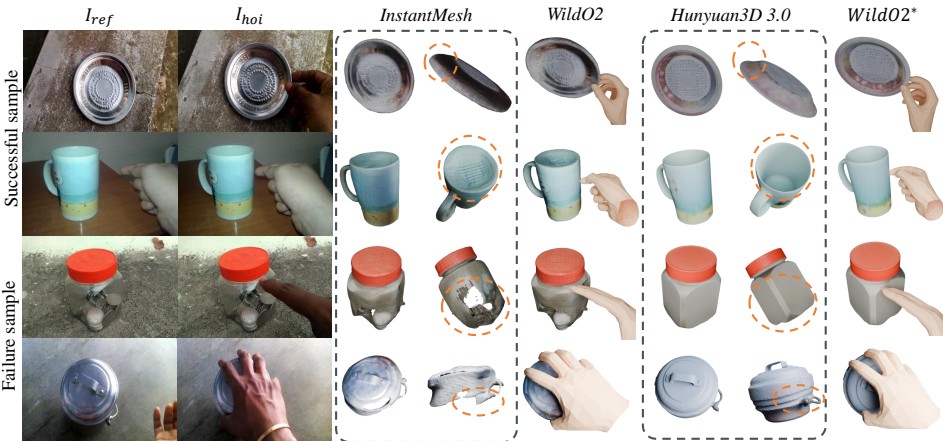

Figure 13: **Qualitative comparison with a stronger reconstruction backbone.** WildO2* denotes results using the Hunyuan3D 3.0 backbone. The first two rows show refinements on previously successful reconstructions, (e.g., sharper details, correct topology) while the bottom two rows demonstrate recovery of previously failed cases. This highlights the pipeline's ability to evolve with advancements in image-to-3D technology.

**Computational and Time Cost Analysis.** On one NVIDIA A40 GPU, our automated steps are efficient: 3D object reconstruction averages 17s per sample with InstantMesh, (ranging from seconds to 3 minutes for other backbones) and our alignment module takes 14s. The subsequent manual verification and refinement, applied to every sample, requires an average of 50 seconds (std = 49s) per person. This commitment to manual oversight is a crucial investment in the final dataset's quality and physical plausibility.

**Human-in-the-Loop Quality Control.** To ensure the highest data fidelity, every sample undergoes mandatory human review. This process is guided by automated quality metrics (CLIP similarity, 2D rendering scores) that assist annotators. Each review results in one of four outcomes:

- Accept: The reconstruction meets all quality standards.

- Refine: Minor inaccuracies are corrected using our dedicated UI (Fig. 14, left).

- Re-generate: A low-quality sample is re-processed with an alternative image-to-3D model.

- Discard: Intractable cases that fail to meet quality standards are removed.

### A.2.2 RECONSTRUCTION ANALYSIS

The primary obstacle is Geometric Reconstruction Failure, where 3D mesh recovery is unsuccessful. Another category, Non-Interactive Cases, is defined for instances where the 2D HOI guide image itself showed no discernible contact. Explicit failures in Pose Estimation account for 2.1%. Finally, the Other Cases category contains various failures that bypassed initial rule-based screening; these include some of the most complex scenarios, such as interactions involving deformable or transparent objects, which fall outside the scope of our current reconstruction method. Details and examples of each failure type are provided in Fig. 14.

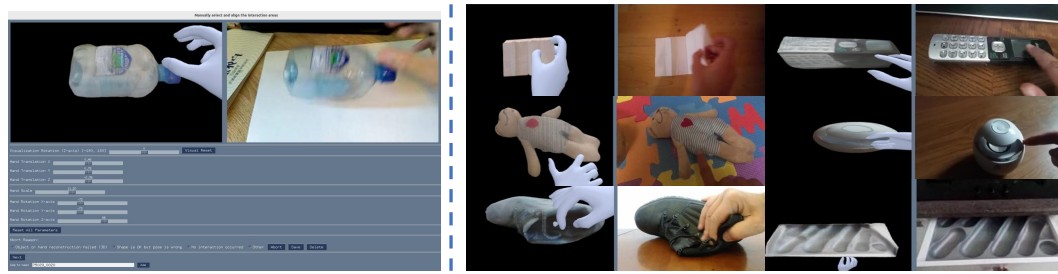

Figure 14: **Examples of reconstruction failures.** (Left) A screenshot of our annotation UI showing a "Non-Interactive" failure, where the reconstructed hand and object do not interact. (Right Top) Examples of "Geometric Reconstruction Failure" for the object and the hand. This was the most common failure category, with object geometry being more challenging to reconstruct than the hand's. (Right Middle) Examples of "Pose Estimation Failure", where the 6DoF pose of the hand or object was incorrectly recovered. (Right Bottom) Examples of "Other Cases". The first example shows a deformable object, and the second shows an object that is part of a larger structure (a drawer).

### A.2.3 SELECTION OF RECONSTRUCTABLE CLIPS FROM LARGE-SCALE IN-THE-WILD VIDEOS

Our data reconstruction and annotation process commences with the Something-Something V2 (SSv2) dataset (Goyal et al., 2017), a large-scale collection of over 220,847 videos. We chose this dataset for several key reasons. First, its vast scale and diversity, resulting from a crowdsourced effort where contributors enact specific verb-noun prompts, provide a rich variety of goal-oriented hand-object interactions. Second, many videos feature close-up shots of the hand-held object, which is advantageous for 3D reconstruction. Critically, the associated Something-Else dataset (Materzynska et al., 2020) furnishes bounding box annotations for hands and objects in approximately 180,049 of these clips, offering an invaluable starting point. However, a primary challenge of SSv2 is its low resolution (typically 240 pixels in height), which necessitates a rigorous filtering pipeline to isolate clips suitable for high-fidelity reconstruction.

To ensure the viability of our reconstruction pipeline, we implemented a multi-stage filtering process to distill the initial 180k annotated clips into a high-quality subset. The screening criteria were designed to simplify the interaction scenario (e.g., single hand, single object) and guarantee sufficient visual quality for reliable 3D modeling. This process effectively removes clips that are ambiguous, occluded, or otherwise intractable. The detailed steps and their impact on the dataset size are summarized in Tab. 6.

Table 6: The multi-stage filtering pipeline for selecting reconstructable clips from the Something-Something V2 dataset. The process starts with clips annotated by Something-Else and progressively refines the selection based on interaction complexity and visual quality.

| Step | Criterion | Description | Clips Remaining |
|---|---|---|---|
| 1 | Initial Annotated Set | Clips from SSv2 with hand and object bounding box annotations. | 180,049 |
| 2 | Single-Hand Interaction | Exclude clips involving multi hands to simplify interaction modeling. | 137,578 |
| 3 | Single-Object Interaction | Retain only clips where a single object is being manipulated. | 82,728 |
| 4 | Minimum Object Size | Discard clips where the object is too small to be reliably reconstructed. | 57,384 |
| 5 | Object Visibility | Exclude clips where the object is too large or moves out of frame. | 12,888 |
| 6 | Frame Stability | Ensure stable pre-interaction and interaction frames can be identified. | 8,551 |

### A.2.4   ACQUISITION OF O2HOI FRAME PAIRS

This section details the automated procedure for extracting an "Object-only to Hand-Object Interaction" (O2HOI) frame pair, denoted as $(I_{\text{ref}}, I_{\text{hoi}})$, from each video clip.

**1.  Mask Generation and Frame Selection Preliminaries.**  For each frame in a given video sequence, we generate object masks ($M_O$) and hand masks ($M_H$) using the SAM2. The interaction period, $\mathcal{T}_{\text{HOI}} = [i_{\text{first}}, i_{\text{last}}]$, is defined as the contiguous block of frames where the Intersection over Union (IoU) between the object mask and the hand mask, each expanded by several pixels, exceeds a predefined threshold. The reference frame, $I_{\text{ref}}$, is selected as the object-only frame temporally closest to this interaction period.

**2. Optimal HOI Frame Selection.**  The primary objective is to select an index $i_{\text{hoi}}$ from $\mathcal{T}_{\text{HOI}}$ such that the object's pose in frame $I_{\text{hoi}}$ has undergone minimal change relative to its pose in $I_{\text{ref}}$. We achieve this by estimating the 2D affine transformation between the object in the reference frame and in every candidate frame within $\mathcal{T}_{\text{HOI}}$. This estimation utilizes the RoMa dense feature matcher (Edstedt et al., 2024). The detailed steps for comparing a candidate frame $I_t$ (where $t \in \mathcal{T}_{\text{HOI}}$) to the reference frame $I_{\text{ref}}$ are as follows:

- Feature Matching: For each pair $(I_{\text{ref}}, I_t)$, we extract robust keypoint correspondences $(\mathbf{k}_{\text{ref}}, \mathbf{k}_t)$ within their respective object masks, $M_{\text{ref}}^O$ and $M_t^O$.

- Transformation Estimation: The keypoints are used to robustly estimate an affine transformation matrix $\mathbf{A}_t$ using RANSAC.

- Rotation and IoU Calculation: The rotation angle sequence $\theta_t$ and IoU metrics are computed as follows:

$$R_S = \mathbf{A}_t[0:2, 0:2], \quad U, S, V^\top = \text{SVD}(R_S), \quad R = UV^\top, \tag{8}$$

$$\theta_t = \arctan 2(R_{21}, R_{11}) \cdot \frac{180}{\pi}, \quad \text{IoU}_t = \frac{|\text{warpAffine}(M_{\text{ref}}^O, \mathbf{A}_t) \cap M_t|}{|\text{warpAffine}(M_{\text{ref}}^O, \mathbf{A}_t) \cup M_t|}. \tag{9}$$

- Interaction Frame Selection: with IoU gradient $\Delta\text{IoU}_t$. The selected HOI frame index $i_{\text{hoi}}$ is determined by:

$$i_{\text{hoi}} = \begin{cases} \arg\min_{t \in \mathcal{T}_{\text{HOI}}} |\theta_t|, & \text{if } \min_{t \in \mathcal{T}_{\text{HOI}}} |\theta_t| > \text{MAX\_MIN\_ANGLE} \\ \arg\min_{t \in \mathcal{S}_{\text{stable}}} |t - i_{\text{max}}|, & \text{if } \max_{t \in \mathcal{T}_{\text{HOI}}} |\theta_t| < \text{MIN\_MAX\_ANGLE} \\ \arg\min_{\substack{t \in \mathcal{T}_{\text{HOI}} \\ |\theta_t| < \text{MAX\_MIN\_ANGLE}}} |t - i_{\text{max}}|, & \text{otherwise} \end{cases} \tag{10}$$

where $\mathcal{S}_{\text{stable}} = \{t \in \mathcal{T}_{\text{HOI}} \mid |\Delta\text{IoU}_t| < \text{DT\_IOU\_THRES}\}$, and MAX\_MIN\_ANGLE=1, MIN\_MAX\_ANGLE=5

- Inpainting Mask Generation: In summary, we obtain the O2HOI frame pair $(i_{\text{ref}}, i_{\text{hoi}})$ and the corresponding inpainting mask for subsequent processing:

$$M_{\text{inpaint}} = \text{warpAffine}(M_{\text{ref}}^O, \mathbf{A}_{\text{hoi}}). \tag{11}$$

This procedure ensures accurate spatial alignment between the object-only and interaction frames, facilitating efficient and reliable inpainting mask generation for downstream tasks.

A.2.5 ESTIMATION OF OBJECT ELEVATIONS

The backbone of InstantMesh(Xu et al., 2024), Zero123(Liu et al., 2023a), generates novel views of objects; however, the predicted elevation angles may not precisely align with the control signals. To address this, we employ a rendering-based matching strategy to estimate the optimal elevation angle for the reconstructed 3D object mesh. Specifically, we render the mesh from candidate elevations and select the angle whose 2D projection best matches the HOI frame, providing a reliable initialization for subsequent optimization. The search is performed in two stages: (a) a coarse search over the range $[-40°, 60°]$ with 11 sampled angles, and (b) a fine search within $[\text{best} - 10°, \text{best} + 10°]$ using 21 samples. The selection criterion is the point-wise mean squared error (MSE) between the rendered projection and the aligned HOI frame.

$$\text{elevation} = \arg\min_{\theta \in \Theta} \frac{1}{N} \sum_{i=1}^{N} (I_{\text{ref}}[i] - I_{\text{rendered}}(\theta)[i])^2. \tag{12}$$

We employ the CLIP-similarity metric to automatically assess the reconstruction quality of image-to-3D methods. This enables efficient and coarse filtering of clips, ensuring that only those with high-quality and reliable 3D reconstructions are selected for subsequent processing and annotation.

A.2.6 CONTACT MAP COMPUTATION

Given an object point cloud $\mathbf{O} \in \mathbb{R}^{N_o \times 3}$ and a hand point cloud $\mathbf{H} \in \mathbb{R}^{N_h \times 3}$, we propose a robust four-stage contact map computation algorithm:

**Stage 1: Bidirectional Nearest Neighbor Voting.** For each point, we compute the nearest neighbor in the opposite set and accumulate votes:

$$\text{vote}_o(i) = \sum_j \mathbf{1}[\text{NN}(h_j) = o_i], \quad \text{vote}_h(j) = \sum_i \mathbf{1}[\text{NN}(o_i) = h_j].$$

**Stage 2: Candidate Selection.** High-frequency candidate points are selected based on the upper quantile $\alpha$ of the vote distribution:

$$\mathcal{C}_o = \{i \mid \text{vote}_o(i) \geq Q_{1-\alpha}(\text{vote}_o)\}.$$

**Stage 3: Distance Validation.** Core contact points are further filtered by requiring their hand-object distance $d_i$ to be below both a quantile threshold $\beta$ and an absolute threshold $\epsilon$:

$$\mathcal{S}_o = \{i \in \mathcal{C}_o \mid d_i < \min(Q_\beta(d), \epsilon)\}.$$

**Stage 4: Region Expansion.** The contact region is expanded by including points within a radius $\gamma \cdot \bar{d}$ of any core contact point, where $\bar{d}$ is the mean $k$-nearest neighbor distance:

$$\mathcal{M}_o = \left\{i \mid \min_{s \in \mathcal{S}_o} \|o_i - o_s\|_2 \leq \gamma \cdot \bar{d}\right\}.$$

A.2.7 WILDO2 DSCS PROMPT

```
[Structured Output Protocol]
As a hand-object interaction analyzer, generate JSON strictly like this:
{
    "obj_category": "cylindrical",
    "general": "grip [obj_category]",
    "physical": "Apply [hand_contact] to establish stable three-point contact with
        [obj_contact] of [obj_category] while other fingers form loose sphere.",
    "hand_contact": ["thumb pad", "pinky nail", "palm"],
    "obj_contact": "body"
}

[Key Requirements]
1. Mandatory placeholders: [hand_contact],[obj_contact],[obj_category]
2. Physical Field Rules:
```

```
    – Use ONLY [hand_contact] placeholder – ABSOLUTELY NO explicit contact point
        enumeration
    – must describe contact areas, interaction methods, and force levels
        considering object functionality
3. hand_contact is where interaction is most likely to occur.Input hand_contact
    may contain sensing errors from point cloud analysis. Valid hand_contact
    options (17 total):['thumb pad','index pad','middle pad','ring pad','pinky
    pad','thumb nail','index nail','middle nail','ring nail','pinky nail','thumb
    knuckle','index knuckle','middle knuckle','ring knuckle','pinky
    knuckle','palm','back of palm',]
4. ABSOLUTELY NO PREAMBLE/FOOTNOTES, only RAW JSON output

Current Input Interaction Description: "hand_contact": xxx, "general_intention":
    xxx.
```

### A.2.8 DATASET ANALYSIS

For data integration purposes, Tab. 7 outlines the mapping used to align labels from the Something-Something dataset with our defined action groups. A comprehensive comparison with existing hand-object interaction datasets is presented in Tab. 8. We further analyze the statistical properties of our collected objects, including the scale distribution and its relationship with action categories, as visualized in Fig. 15.

Table 7: The correspondence between the class labels of the Something-Something dataset and the action groups defined in this study.

| Action Group | Class Label |
|---|---|
| **Picking** | Pretending to pick something up |
| | Picking something up |
| **Pushing** | Pushing something from right to left |
| | Pushing something from left to right |
| | Pushing something so that it almost falls off but doesn't |
| | Pushing something so that it slightly moves |
| | Pushing something so it spins |
| | Pushing something so that it falls off the table |
| | Pushing something off of something |
| | Pushing something onto something |
| | Pushing something with something |
| | Something colliding with something and both are being deflected |
| **Poking** | Poking something so that it falls over |
| | Poking a stack of something so the stack collapses |
| | Poking something so it slightly moves |
| | Poking something so lightly that it doesn't or almost doesn't move |
| | Pretending to poke something |
| | Poking something so that it spins around |
| | Poking a stack of something without the stack collapsing |
| | Poking a hole into something soft |
| | Poking a hole into some substance |
| **Lifting** | Lifting up one end of something without letting it drop down |
| | Lifting something up completely without letting it drop down |
| | Lifting up one end of something, then letting it drop down |
| | Lifting something up completely, then letting it drop down |
| | Lifting something with something on it |

| | |
|---|---|
| | Lifting a surface with something on it but not enough for it to slide down |
| **Moving** | Moving something down |
| | Moving something towards the camera |
| | Moving something away from the camera |
| | Moving something across a surface without it falling down |
| | Moving something up |
| | Moving something across a surface until it falls down |
| | Moving part of something |
| **Squeezing** | Pretending to squeeze something |
| | Squeezing something |
| **Taking** | Taking one of many similar things on the table |
| | Pretending to take something from somewhere |
| | Pretending to take something out of something |
| | Taking something from somewhere |
| **Touching** | Touching (without moving) part of something |
| **Opening** | Opening something |
| | Pretending to open something without actually opening it |
| **Laying** | Laying something on the table on its side, not upright |
| **Turning** | Pretending to turn something upside down |
| | Turning something upside down |
| **Rolling** | Rolling something on a flat surface |
| | Letting something roll along a flat surface |
| | Letting something roll down a slanted surface |
| **Spinning** | Spinning something that quickly stops spinning |
| | Spinning something so it continues spinning |
| **Pulling** | Pulling something from right to left |
| | Pulling something from left to right |
| **Throwing** | Throwing something |
| | Throwing something onto a surface |
| | Throwing something in the air and letting it fall |
| **Putting** | Putting something on a surface |
| | Putting something upright on the table |
| | Putting something on a flat surface without letting it roll |
| | Putting something onto a slanted surface but it doesn't glide down |
| | Putting something into something |
| | Putting something that can't roll onto a slanted surface, so it slides down |
| | Pretending to put something behind something |
| | Putting something that cannot actually stand upright upright on the table, so it falls on its side |
| | Putting something similar to other things that are already on the table |
| **Falling** | Something falling like a feather or paper |
| | Something falling like a rock |
| **Bending** | Bending something so that it deforms |
| **Tipping** | Tipping something over |
| | Tipping something with something in it over, so something in it falls out |
| **Closing** | Pretending to close something without actually closing it |
| | Closing something |
| **Twisting** | Twisting something |

| | Pretending or trying and failing to twist something |
|---|---|
| **Holding** | Holding something |
| **Piling** | Piling something up |
| **Showing** | Showing that something is empty |
| | Showing something to the camera |
| **Sprinkling** | Pretending to sprinkle air onto something |
| **Tearing** | Tearing something just a little bit |
| | Pretending to be tearing something that is not tearable |
| | Tearing something into two pieces |
| **Folding** | Folding something |
| **Unfolding** | Unfolding something |
| **Attaching** | Attaching something to something |
| **Tilting** | Tilting something with something on it slightly so it doesn't fall down |
| | Tilting something with something on it until it falls off |
| **Stacking** | Stacking number of something |
| **Covering** | Covering something with something |
| **Spreading** | Pretending to spread air onto something |
| **Dropping** | Dropping something onto something |
| **Wiping** | Pretending or failing to wipe something off of something |

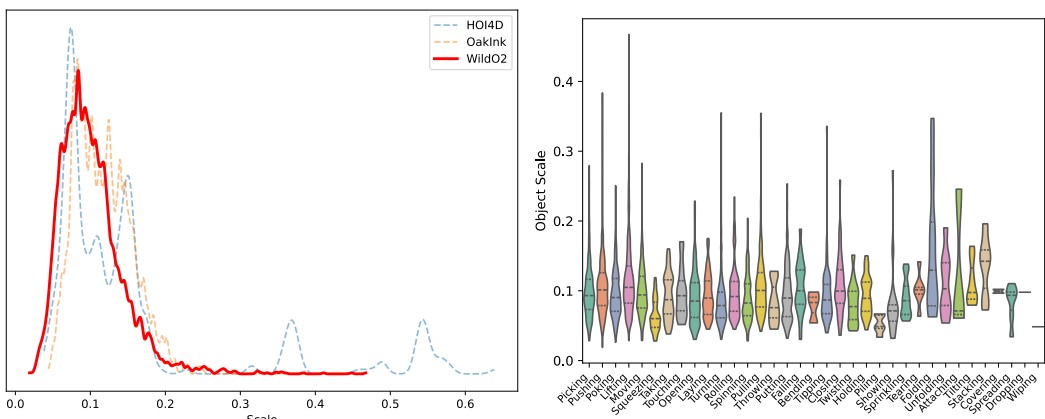

Figure 15: The left figure is KDE curves of object scales in different datasets. Our dataset, collected without object category bias, shows a more uniform scale distribution., while the right figure depicts the object size distribution across different verb categories. Actions like "squeezing" typically involve smaller objects, whereas "folding" tends to be associated with larger ones.

## A.3  DETAILS OF TOUCH

### A.3.1  IMPLEMENTATION DETAILS

**Condition Injection Transformer Architecture**. Our diffusion model and refiner network $f_{\text{refiner}}$ are both built upon an 8-layer, 4-head Transformer architecture. This architecture features a latent dimension of 512, a feed-forward network size of 1024, the GELU activation function, and a dropout rate of 0.1. To handle the variable number of input points for hands and objects, we designed an Adaptive Feature Selector, which employs a multi-head attention mechanism to aggregate features from the input point clouds into a fixed-size representation (64 for the hand and 128 for the object). **Text Encoder**. To facilitate effective feature alignment and representation learning, we designed and implemented customized encoder architectures tailored to different sources of textual features.

| Dataset | Data Collection | | Dataset Scale | | | Dataset Anno. | |
|---------|-----------------|---|---------------|---|---|---------------|---|
| | Environmnet | Authentic | Obj Cate. | Obj Inst. | Clips/Frames | Act. Cate. | Contact |
| HO-3D | lab | ✓ | 10 | 10 | 27/78k | none | ✗ |
| obman | syn. | ✗ | 8 | 2.7k | -/154K | none | ✗ |
| HOI4D | lab | ✓ | 16 | 800 | 4K/2.4M | 54 | ✗ |
| Oakink* | lab/syn. | ✓/✗ | 34/34 | 100/1.7k | 793/230k | 5 | ✗ |
| ContactPose | lab | ✓ | 25 | 25 | 2.3k/3M | 25 | ✓ |
| GRAB | lab | ✓ | 37 | 51 | 1.3k/1.6M | 4 | ✗ |
| *Ours* | wild | ✓ | 403 | 4.4k | 4.4k/- | 92 | ✓ |

Table 8: **Comparison with existing hand-object interaction datasets:** HO-3D (Hampali et al., 2020), obman (Hasson et al., 2019), HOI4D (Liu et al., 2022), Oakink* (Yang et al., 2022), ContactPose (Brahmbhatt et al., 2020), GRAB (Taheri et al., 2020). "Authentic" indicates whether the hand-object interaction is genuine human behavior. The Oakink* dataset comprises two parts: real data collected in laboratory settings (img set) and synthetic data (shape set).

For the 4096-dimensional features from the large language model Qwen-7B, we employed a non-linear feature adapter. This module, inspired by established practices in cross-modal alignment, utilizes a Multi-Layer Perceptron (MLP) incorporating GELU activations and Layer Normalization to project the high-dimensional features into a unified 512-dimensional latent space. For the 768-dimensional token-level sequence features from models such as BERT and MPNet, we diverged from the conventional mean-pooling strategy. Instead, we implemented an attention-based pooling mechanism.

**Training**. The diffusion model is trained for 1000 epochs using the Adam optimizer with a learning rate of 1e-4 and a batch size of 128. Subsequently, the refiner network is trained using a distinct set of loss weights tailored for physical plausibility. During this phase, the parameters of the diffusion model are kept frozen.

**Refinement**. At inference time, following a single forward pass through the refiner network, we perform Test-Time Adaptation (TTA) on the generated pose. This process directly optimizes the pose parameters for 500 iterations with a learning rate of 1e-2. The loss weight coefficients used during all training and refinement stages are detailed in Tab. 9.

Table 9: Weight coefficients for each loss term during the training and refinement stages. - indicates that the loss term is not applicable to the corresponding stage.

| Parameter | Diffusion Model | Refiner Network | Description |
|-----------|-----------------|-----------------|-------------|
| $\lambda_{\text{simple}}$ | 1.0 | 5.0 | Base L2 loss |
| $\lambda_{\text{dmap}}$ | 0.1 | - | Hand-object distance map loss |
| $\lambda_{\text{global}}$ | 0.1 | 0.1 | Hand global pose loss |
| $\lambda_{\text{pene}}$ | - | 100.0 | Hand-object penetration loss |
| $\lambda_{\text{contact}}$ | - | 100.0 | Hand-object contact loss |
| $\lambda_{\text{cyc}}$ | - | 10.0 | Cycle-consistency loss |
| $\lambda_{\text{self}}$ | - | 10000.0 | Hand self-penetration loss |
| $\lambda_{\text{anatomy}}$ | - | 0.1 | Anatomical plausibility loss |

### A.3.2 EXPERIMENT SETTINGS

**Training Details.** We implement our model using Accelerate and train it on 8 NVIDIA 4090D GPUs with a batch size of 128. We first define 17 detailed hand parts (e.g., pad, nail, knuckle for each finger, palmar, dorsal). Due to the long-tailed distribution of their contact frequencies, we

aggregate these parts into 7 semantically coherent categories (pad of each finger, palmar and dorsal). The contact state for any interaction is then encoded as a 7-bit binary label, where each bit represents the contact status (1 for contact, 0 for non-contact) of one category. To create a balanced training set, we perform resampling based on these unique 7-bit labels.

**Evaluation Metrics**

- **Physical Plausibility.** We employ three metrics to assess the physical plausibility of the generated hand-object interactions: (1) Mean Per-Vertex Position Error (MPVPE, mm), which computes the average L2 distance between the predicted hand mesh $\hat{\mathbf{H}}$ and the ground truth $\mathbf{H}$; (2) Penetration Depth (PD, cm), measuring the maximum depth of hand vertices penetrating the object surface; (3) Penetration Volume (PV, cm$^3$), quantifying the volumetric intersection by voxelizing the object mesh and calculating the volume within the hand surface.

- **Contact Accuracy.** Contact accuracy is measured by comparing the predicted contact map $\mathbf{C}_H$ with the ground truth $\mathbf{C}_H^*$ using Intersection over Union (IoU) and F1 score.

- **HOI Diversity.** Following (Liu et al., 2023b), we assess diversity by clustering generated grasps into 20 clusters using K-means, and report the entropy of cluster assignments and the average cluster size.

- **Semantic Consistency.** Semantic consistency is evaluated by: (1) P-FID, the Fréchet Inception Distance between point clouds of predicted and ground truth hand meshes, using a pre-trained feature extractor (Nichol et al., 2022); (2) VLM assisted evaluation, where rendered hand-object interactions are scored for semantic alignment with input captions (0-10 scale); (3) Perceptual Score (PS), the mean rating from 10 volunteers (0-10 scale), reflecting the naturalness and semantic consistency of generated grasps.

## A.4  USE OF LLMS

We utilized Large Language Models (LLMs) and Vision-Language Models (VLMs) to assist in various stages of this research. Specifically, their use can be broken down as follows:

- Dataset Annotation: During the dataset generation phase, we employed both LLMs and VLMs to automatically generate descriptive text annotations.

- Model Architecture: Our proposed model architecture directly incorporates prior knowledge from a pre-trained LLM (Qwen-7B) to enhance its reasoning and generation capabilities.

- Evaluation: We used a VLM as part of our automated pipeline for the quantitative evaluation of our model's generated outputs.

- Manuscript Preparation: We used an LLM-based tool for proofreading to help.

