# OpenReview forum: "TOUCH: Text-guided Controllable Generation of Free-Form Hand-Object Interactions"
_ICLR.cc/2026/Conference — ICLR 2026 Poster_

### Official Review · Reviewer_yAMZ · 2025-10-30

**Soundness:** 4
**Presentation:** 4
**Contribution:** 4
**Rating:** 8
**Confidence:** 5

**Summary:**

This paper tackles free-form HOI generation, pushing the field beyond the standard grasp-centric paradigm. The authors introduce two main contributions: WildO2, a novel 3D HOI dataset reconstructed from in-the-wild videos, and TOUCH, a three-stage diffusion framework designed to synthesize these diverse interactions from fine-grained text prompts. The work is solid, and the results are impressive.

**Strengths:**

1.	The formulation of the "Free-Form HOI" task is a forward-looking contribution. It moves the community beyond the well-trodden "grasping" paradigm toward more realistic and diverse interactions.
2.	The proposed data pipeline for reconstructing 3D HOIs from monocular videos is effective. Creating an in-the-wild 3D dataset like WildO2 is a valuable asset for the community.
3.	The TOUCH framework is technically sound. Its three-stage approach is a logical decomposition of the problem, and the coarse-to-fine conditioning is an effective strategy for fine-grained text control.

**Weaknesses:**

1.	The dataset pipeline's quality is naturally capped by its upstream components (e.g., image-to-3D models, SAM2). Existing 3D generation methods are less used for in-the-wild, low-resolution generation, and a discussion on how to improve the accuracy of the dataset synthesis method in the future would be beneficial.
2.	WildO2 excels in interaction diversity. However, its absolute scale is understandably smaller than that of massive lab datasets (e.g., Gigahands).
3.	The method's generalizability needs more validation. Adding experiments on other datasets/domains would strengthen the paper's claims, e.g., qualitative results on OakInk and tests on open-set object or CAD models from sources like Objaverse.
4.	The 4/4 layer split for coarse-to-fine conditioning appears empirical. An ablation study is needed to justify this specific architectural choice against other alternatives (e.g., 2/6).

**Questions:**

1.	The reconstruction failure analysis for the dataset is great. Could you also show some typical generation failures of the TOUCH model itself w.r.t. certain objects or text prompts?
2.	What is the inference speed of the full pipeline? How much overhead does the TTA add, and is it critical for performance?
3.	There is an incomplete citation for Ye et al. (L698) and a missing space in 'WildO2that' (L99).

---

> ### Author Response · Authors · 2025-11-25
> **Responses to yAMZ**
>
> ### Responses to Weaknesses
>
> ---
>
> 1. **On Future Improvements for the Pipeline:** We agree and wish to emphasize that the pipeline's quality is intrinsically linked to its upstream components, which is a key aspect of its evolvable design. Future accuracy improvements can be achieved through two clear paths: 1) Integrating more advanced image-to-3D models as they become available, such as recent methods like SAM3D that enable scene-level reconstruction. 2) Leveraging higher-resolution data sources (e.g., from datasets like Ego4D) to overcome the quality ceiling imposed by the low-resolution (<420p) clips used in our current collection.
> 2. **On the Scalability of WildO2:** Our work deliberately prioritizes interaction diversity and in-the-wild realism—qualities that are difficult to capture in controlled lab settings. The quantitative comparison with GigaHands below highlights this distinction.
>
> |  | Clips | **Object Instants** | **Text Annotations**. |
> | --- | --- | --- | --- |
> | GigaHands | **14k** | 417 | **84k** |
> | WildO2 | 4.4k | **4,414** | 9k |
>
> While GigaHands is larger in volume, its reliance on costly motion capture systems limits its scalability in object diversity. Therefore, our primary contribution is the scalable pipeline itself, which provides a blueprint for efficiently expanding the dataset to rival the scale of lab-based collections while retaining its unique diversity.
>
> 3. **Out-of-domain**:To further validate out-of-domain generalization, we tested our method on a set of portable object CAD models randomly sampled from Objaverse. We paired these with diverse, LLM-generated captions. As visualized in [***link***](https://anonymous.4open.science/w/hoi123touch-6121/#Out-of-domain) (main manuscript Figure 7, L461), our approach successfully produces plausible interaction poses for these novel CAD models, demonstrating its strong generalization ability.
> 4. **On Ablation Study for Layer Split:** We conducted the suggested ablation study to justify this architectural choice. The results, summarized below, reveal two key findings. First, our 4/4 configuration achieves the best overall performance, confirming that a balanced injection strategy is optimal. Second, we observe that the 2/6 split (favoring late injection of fine-grained information) significantly outperforms the 6/2 split (favoring early injection of coarse information). This indicates that for free-form HOI synthesis, providing fine-grained conditional information is more critical.
>
> | metrics | P-IOU ↑ | P-F1 ↑ | MPVPE ↓ | PD ↓ | PV ↓ |
> | --- | --- | --- | --- | --- | --- |
> | 0/8 (✗ mul) | 0.766 | 0.836 | 2.97 | 0.950 | 2.55 |
> | 2/6 | 0.767 | 0.839 | 2.97 | 0.942 | **2.52** |
> | 4/4 | **0.776** | **0.844** | **2.89** | **0.932** | 2.67 |
> | 6/2 | 0.650 | 0.747 | 3.64 | 1.026 | 4.46 |
>
> ### Responses to Questions
>
> 1. **Typical generation failures of TOUCH:** Thank you for this question. A qualitative analysis of typical failure cases is provided in [***link***](https://anonymous.4open.science/w/hoi123touch-6121/#TOUCH-Failure-Cases) (Appendix A.2.3, L908). Key failure modes include: (a) Grasp Bias: A bias towards grasping poses, even when the text prompt lacks corresponding cues. (b) Incorrect palm orientation. (c) Inaccurate contact region guidance. (d) Severe fingertip penetration.
> 2.  **Inference speed of the full pipeline:** The inference pipeline consists of three stages, with speeds detailed below. While TTA is the most time-consuming step, we find it indispensable for achieving high-fidelity results. Our work prioritizes generation quality and controllability, and we believe the significant performance gain justifies the TTA overhead.
>
> | Stage | DDIM |  initial refinement | TTA |
> | --- | --- | --- | --- |
> | **Time** | ~6s | ~0.3s | ~107s for 250 iters |
>
> 3. Thank you for pointing these out. We have corrected the incomplete citation and the typo in the revised manuscript.

---

> > ### Comment · Reviewer_yAMZ · 2025-11-27
> > **good paper and tend to accept**
> >
> > The authors addressed my concerns and I believe this is a good paper to be published.

---

> > > ### Author Response · Authors · 2025-11-27
> > > **Grateful for your time and careful review**
> > >
> > > Thank you for your appreciation of our work and for your careful and thorough review！

---

### Official Review · Reviewer_TWqj · 2025-11-01

**Soundness:** 3
**Presentation:** 3
**Contribution:** 3
**Rating:** 8
**Confidence:** 4

**Summary:**

The paper addresses a critical data bottleneck in the hand–object interaction (HOI) domain — the lack of high-quality 3D datasets capturing free-form, non-grasping interactions. While existing datasets focus almost exclusively on structured grasping scenarios collected in laboratory settings, this work proposes WildO2, an in-the-wild 3D HOI dataset that covers diverse everyday manipulations such as pushing, poking, turning, and rotating. WildO2 is automatically constructed from internet videos using an object-only to interaction (O2HOI) frame pairing pipeline, followed by multi-stage 3D reconstruction, contact optimization, and text-based semantic annotation via vision–language models.

Building on this dataset, the authors introduce TOUCH, a three-stage text-guided framework for controllable HOI generation. TOUCH integrates (1) explicit contact map prediction, (2) a multi-level conditioned diffusion model that fuses coarse-to-fine text and geometric cues, and (3) a physical refinement module ensuring realistic contact and alignment. Experiments show that TOUCH generates diverse, semantically aligned, and physically plausible free-form interactions, outperforming prior baselines (e.g., ContactGen, Text2HOI) in contact accuracy, plausibility, and diversity metrics.

**Strengths:**

The paper’s WildO2 dataset is a major technical contribution, featuring an well-designed and automated data generation pipeline. This pipeline successfully integrate multi-stage object–hand reconstruction, camera alignment, and physical contact refinement, resulting in high-quality 3D annotations and realistic HOI samples. The inclusion of Descriptive Synthetic Captions (DSCs), generated and verified through vision–language models, is particularly valuable for enabling text-guided interaction synthesis tasks.

The dataset specifically targets free-form, non-grasping hand–object interactions—a type of everyday manipulation that is pervasive in the real world but consistently overlooked in prior HOI datasets, which mostly emphasize stable grasping or object holding.  the work fills a clear research gap and opens new possibilities for studying intent-driven, semantically controllable HOI generation in both computer vision and embodied AI

the proposed TOUCH framework (contact → pose → refinement) follows a fairly typical architecture within current interaction synthesis pipelines, it is well-implemented and well-validated through both quantitative and qualitative experiments. Its role here effectively complements the dataset.

**Weaknesses:**

The WildO2 dataset primarily focuses on rigid objects, while articulated or deformable objects (e.g., clothes, plastic bags, napkins) are absent. These categories are often the most likely to trigger free-form and dynamic hand–object interactions in everyday activities. Although using rigid objects is acceptable for building an initial benchmark, this omission limits the dataset’s ability to fully capture the spectrum of natural, unconstrained human–object interactions.

Despite the paper’s aim to model free-form interactions, the proposed TOUCH framework largely inherits design principles from grasp-based synthesis—treating contact as a quasi-static grasping state. While this formulation is reasonable for static contact modeling, free-form interactions are inherently motion-centric, and thus would benefit from a dynamic or sequence-level synthesis perspective rather than purely static pose generation.

The use of the term data generation sec 3.2 may be somewhat misleading, as the proposed pipeline mainly performs 3D reconstruction and alignment rather than generative modeling. Although the inclusion of LLM-generated Descriptive Synthetic Captions (DSCs) introduces a generative component, the overall process is better described as a data reconstruction or annotation pipeline to avoid misleading .

**Questions:**

What is the average processing time per frame in the data generation pipeline for WildO2? It would be helpful to know the computational cost and scalability of the proposed reconstruction and alignment procedure.

 In the WildO2 dataset, how were the object categories and action types selected? Do the defined free-form action labels correspond to common patterns of real-world human activity, or were they primarily derived from the source video dataset?

---

> ### Author Response · Authors · 2025-11-25
> **Responses to TWqj**
>
> Thank you for your valuable feedback. Your comments are highly constructive, and we will revise the paper accordingly.
>
> ---
>
> ### Responses to Weaknesses:
>
> 1. **On Deformable Objects:** We thank the reviewer for this insightful point. We clarify that while our WildO2 dataset includes deformable objects (e.g., clothes, plastic bags, napkins, as seen in  [***link***](https://anonymous.4open.science/w/hoi123touch-6121/#deformable_object)), they are represented as static meshes. This is an inherent constraint of single-image 3D reconstruction, which lacks the temporal context needed to capture dynamics.
>
>     However, our framework itself provides a direct and practical methodology for this challenge: representing deformation as a sequence of discrete states. By applying our pipeline across multiple video frames, one can generate a time-series of geometries ($pts_{t_1}, pts_{t_2}, ...$) that effectively approximates the object's dynamic behavior. As underlying image-to-3D and image-to-world technologies continue their expansion toward robust scene-level and in-the-wild reconstruction, this approach is poised to become increasingly powerful.
>
>     Therefore, while the current dataset establishes a robust static benchmark, our framework provides a concrete and actionable blueprint to extend this work toward capturing dynamic, non-rigid HOIs.
>
> 2. **On Static vs. Dynamic Synthesis:** We concur with the reviewer that dynamic synthesis is a key long-term objective. Our work is intentionally focused on the static case because we argue that the foundational challenge of generating a sufficiently diverse and plausible static contact pattern has remained unsolved.
>
>     Our model intentionally moves beyond the restrictive, palmar-focused priors of grasp synthesis. By modeling the entire hand surface and using fine-grained control to avoid mode collapse, TOUCH unlocks the diverse, non-prehensile contact patterns that are essential for true free-form interaction.
>
>     In essence, we tackle the prerequisite "what pose to create" problem, providing the essential and non-trivial building blocks that enable future work to meaningfully address the dynamic "how to sequence them" challenge.
>
> 3. **On the term "data generation":** We accept the reviewer's suggestion. The term “Data Reconstruction Pipeline” more accurately describes our contribution, and we will revise this terminology throughout the manuscript. Thank you for this precise suggestion.
>
> ### Responses to Questions:
>
> ---
>
> 1.  **On Processing Time and Scalability:** Thank you for this practical question. First, we would like to clarify that our pipeline operates on a per-image, not per-frame, basis. On a single NVIDIA A40 GPU, the core automated process is efficient and is completed in **under one minute per sample**. Other lightweight steps, such as SAM-based segmentation and frame matching, are negligible in comparison. This efficiency demonstrates the strong scalability of our approach. A detailed time breakdown is provided below:
>
> | **Step** | InstantMesh | WildO2 Alignment| Manual Verification | HaMeR| Other Recon. Methods |
> | --- | --- | --- | --- | --- | --- |
> | **Avg. Time** | ~17s | ~14s | ~50s (std=49s) | ~1s | 1s to 3min |
>
> 2. **On Category Selection:** Our object categories and action labels are directly inherited from the large-scale Something-Something V2 video dataset, as noted in the main manuscript (L242). This approach ensures that our interaction patterns are grounded in common, real-world human activities and allows us to build upon a well-established benchmark, thereby minimizing selection bias.

---

### Official Review · Reviewer_8tRM · 2025-11-01

**Soundness:** 3
**Presentation:** 3
**Contribution:** 3
**Rating:** 6
**Confidence:** 4

**Summary:**

The paper introduces Free-Form HOI Generation, emphasizing controllable and semantically rich interaction synthesis beyond grasping. In addition, the paper proposes TOUCH, a three-stage framework for text-guided, controllable generation of free-form hand-object interactions (HOI). The multi-level diffusion framework is conditioned on fine-grained text and contact maps, integrating global and local semantic cues for physically plausible synthesis. Finally, the paper introduces WildO2, a large in-the-wild 3D HOI dataset (4.4k interactions, 92 intents, 403 objects) from internet videos via an automated O2HOI reconstruction pipeline. Experiments show the advantage of TOUCH over baselines (ContactGen, Text2HOI) in contact accuracy, plausibility, and semantic alignment.

**Strengths:**

1. The paper addresses free-form HOI generation with fine-grained textual control. The three-stage design effectively combines semantics, geometry, and physics.

2. Comprehensive dataset: WildO2 offers unprecedented diversity, with detailed contact annotations and high-quality reconstructions from in-the-wild videos.

3. Strong quantitative and qualitative performance: improvements over state-of-the-art HOI generation baselines across multiple metrics.

4. Clear ablations and insightful analyses: The impact of contact maps, coarse/fine text, and physical consistency is systematically evaluated.

**Weaknesses:**

1. Static generation limitation: TOUCH focuses on single-frame poses; temporal dynamics (motion continuity, causality) are left for future work.

2. Dataset scale and noise: Although diverse, WildO2 (4.4k samples) remains smaller, and in-the-wild reconstruction errors (≈45% failure rate) suggest potential biases.

3. Comparisons could be expanded: While ContactGen and Text2HOI are solid baselines, comparisons with other text-conditioned 3D diffusion or affordance models (e.g., DiffH2O, Nl2Contact) would strengthen positioning.

4. Ablations on language encoder: The Qwen-7B module shows gains, but results for alternative encoders (e.g., CLIP, BERT) are only briefly summarized. An analysis of semantic faithfulness could be better.

5. Limited discussion on cross-domain generalization: It is unclear how the model generalizes to unseen object categories or out-of-distribution verbs beyond the 92 labeled intents.

**Questions:**

1. How does TOUCH handle ambiguous or conflicting textual intents (e.g., “loosely hold” vs. “grasp tightly”)?

2. What about the generalization to unseen object categories or verbs in WildO2?

3. Could the refinement module be extended to temporal HOI (e.g., multi-frame optimization)?

4. For dataset details, will WildO2 include the intermediate 2D-3D alignment pipeline and failure cases?

5. How does the model behave when the text omits contact information (e.g., only “push the cup”)?

---

> ### Author Response · Authors · 2025-11-25
>
> ### Responses to Weaknesses:
>
> 1. **On Static Generation:** We concur with the reviewer that temporal dynamics is a key long-term objective. Our work focuses on synthesizing diverse static poses, which we consider a fundamental yet unsolved prerequisite. These static poses serve as the building blocks for any dynamic interaction. Therefore, we address the foundational "what poses to create" problem to pave the way for future research on "how to sequence them." This sequential challenge is further compounded by the scarcity of large-scale, in-the-wild 4D interaction data.
> 2. **On Dataset Scale and Quality Control:** Our primary contribution is indeed the novel and scalable data generation pipeline itself, rather than the absolute scale of the current dataset. The ≈45% discard rate is a testament to our rigorous Human-in-the-Loop quality control, which actively filters out low-quality or biased samples. As underlying 3D generation models improve, this success rate will naturally increase, making our validated pipeline an even more powerful tool for future large-scale expansion.
> 3. **On Expanded Comparisons:** Thank you for this valuable suggestion. We have expanded our quantitative evaluation to include both DiffH2O and NL2Contact. Since DiffH2O is a 4D generation method, we adapted its static grasp generation module for our 3D HOI task. As NL2Contact is not open-source, we re-implemented it following the methodology described in their paper. The result can be found in the second point of [Responses to JuZb](https://openreview.net/forum?id=4VW9HVCRw0&noteId=VWOhs5ba1b).
> 4. The qualitative analysis of semantic faithfulness is presented as follows: [***link***](https://anonymous.4open.science/w/hoi123touch-6121/#Semantic-Faithfulness). (main manuscript Figure 8, L482)
> 5. **On Cross-Domain Generalization:** Our model's generalization capability stems from its diversified data and training strategy, which centers on resampling by contact pattern diversity rather than object or action categories. This approach inherently reduces biases towards specific classes. To explicitly validate this, we conducted a new experiment on out-of-distribution objects from Objaverse, paired with LLM-generated captions that mimic our DSC format. As visualized in [***link***](https://anonymous.4open.science/w/hoi123touch-6121/#Out-of-domain) (main manuscript Figure 7, L462), our model successfully generates plausible interactions for these unseen objects, demonstrating its strong generalization capabilities.
>
> ---
>
> ### Responses to Questions:
>
> > 1. **How does TOUCH handle ambiguous intents (e.g., “loosely hold” vs. “grasp tightly”)?**
>
> While our model does not explicitly model physical forces, it learns to interpret such semantic nuances from data-driven priors. Prompts with "tightly/firmly" correlate with larger contact regions in our dataset, whereas "loosely/gently" corresponds to sparser or  more marginal contact. We provide qualitative visualizations in [***link***](https://anonymous.4open.science/w/hoi123touch-6121/#Force-Expression) (main manuscript Figure 9, L487)to illustrate this.
>
> Furthermore, our quantitative analysis corroborates this observation. The table below shows that interactions conditioned on "firm/tight" prompts have, on average, a 25% larger contact area than those conditioned on "gentle/loose" prompts.
>
> |  | **Hand Contact Area (%)** | **Object Contact Area (%)** |
> | --- | --- | --- |
> | gentle/loose | 4.87% | 3.21% |
> | firm/tight | 5.97% | 4.02% |
>
> > 2. **Generalization to unseen object categories or verbs in WildO2?**
> >
>
> Generalization to unseen object categories is enabled by the upstream image-to-3D model, which can reconstruct novel objects. Our action labels are directly inherited from the comprehensive Something-Something v2 dataset, ensuring a broad and representative set of real-world interactions.
>
> > 3. **Could the refinement module be extended to temporal HOI?**
> >
> - Architectural Readiness: Our refinement module shares the same core Conditional Injection Module (CIM) as our multi-level diffusion generator, differing only in its use of physics-based losses for fine-tuning. This architectural parallel means it can be readily updated for sequential data by incorporating standard techniques like time embeddings.
> - TTA for Temporal Coherence: TTA can be leveraged to ensure temporal consistency. This is achieved by simply augmenting its optimization objective with a consistency loss term, which penalizes large kinematic changes between adjacent frames.

---

> ### Author Response · Authors · 2025-11-25
>
> > 4. **Will WildO2 include the intermediate 2D-3D pipeline and failure cases?**
> >
>
>  Yes. Our pipeline integrates a rigorous Human-in-the-Loop Quality Control process. Every sample undergoes mandatory human review, guided by automated metrics. Annotators can choose to accept high-quality samples, refine minor inaccuracies using a dedicated UI, re-generate with an alternative model, or discard intractable cases. A detailed analysis and visualization of typical failure cases are provided in  [***link***](https://anonymous.4open.science/w/hoi123touch-6121/#recon_fail) (Appendix A.3.2, L1012).
>
> > 5. **How does the model behave when the text omits contact information?**
> >
>
> This is an excellent question that is directly addressed by our ablation study in main manuscript Table 2 (L439). In this study, the experiment ✗ $T_{DSC}$ represents conditioning only with SSCs (Short Synthetic Captions, meaning "the prompt omits contact information").
>
> |  | P-IOU ↑ | P-F1 ↑ | MPVPE ↓ | PD ↓ | PV ↓ | P-FID ↓ |
> | --- | --- | --- | --- | --- | --- | --- |
> | ✗ $T_{DSC}$ | 0.698 | 0.784 | 3.02 | 1.119 | 5.28 | 6.09 |
> | Ours(✗ TTA) | **0.728** |**0.805**| **3.00** | **1.093** | **4.82** |**4.84** |
>
> As [***link***](https://anonymous.4open.science/w/hoi123touch-6121/#dsc) (Appendix A.2.1, L880) demonstrated, when fine-grained contact directives are omitted (e.g., using only "put tea bag on a surface"), the model tends to collapse to common grasping patterns.

---

### Official Review · Reviewer_JuZb · 2025-11-01

**Soundness:** 3
**Presentation:** 2
**Contribution:** 2
**Rating:** 6
**Confidence:** 4

**Summary:**

The paper reframes HOI generation from grasp-centric scenarios to free-form interactions and introduces WildO2, an in-the-wild 3D HOI dataset constructed via an O2HOI pairing-and-reconstruction pipeline. It proposes TOUCH, a three-stage framework: (i) text- and geometry-conditioned CVAEs that predict hand/object contact maps; (ii) a multi-stage conditional diffusion model that injects global cues with coarse SSC text early and local geometry with fine DSC text late; and (iii) a lightweight physics-constrained refiner with cycle-consistent contact to correct global pose and sharpen local contacts. The system aims to produce controllable, diverse, and physically plausible interactions beyond grasping.

**Strengths:**

1. The paper contributes a relatively comprehensive dataset to the community and provides useful dataset statistics and analyses in the supp.
2. The method’s explicit prediction of contact regions, coupled with a coarse-to-fine conditioning schedule, is conceptually sound and well aligned with the goal of improving both global plausibility and local contact fidelity.

**Weaknesses:**

1. The approach (and the dataset pipeline) relies on one-image-to-3D reconstructions, especially during TTA. Inaccurate object reconstruction can propagate to and bias the estimated hand pose. The paper should analyze or mitigate this dependency—for example, via robustness studies under controlled reconstruction noise, uncertainty-aware weighting, or comparisons with stronger/alternative reconstruction backbones.
2. While modeling dorsal-side contact is interesting, the paper does not clearly articulate advantages over prior grasp-generation methods such as SemGrasp, which also specifies finger contacts and applies TTA for post-processing. A direct comparison—quantitative and qualitative—under matched prompts and settings would better substantiate the claimed benefits.

**Questions:**

The following questions are based on the weaknesses discussed above; please refer to that section.

---

> ### Author Response · Authors · 2025-11-25
> **Responses to JuZb**
>
> Dear Reviewer, thank you for your valuable feedback. Our responses to your questions are as follows:
>
> ---
>
> ### 1. On the Dependency on Image-to-3D Reconstruction
>
> Thank you for this important question. We would like to clarify a crucial distinction: the image-to-3D technology is utilized exclusively for our dataset creation pipeline and is not part of the TOUCH model's inference TTA process. During inference, TOUCH takes a given 3D object as input, and TTA optimizes the hand pose on that fixed object.
>
> Nevertheless, to address your suggestion, we have supplemented our paper with a comparative analysis using a stronger, closed-source backbone (Hunyuan3D 3.0). This experiment demonstrates that our data pipeline directly benefits from more advanced upstream models, improving both the quality of successful reconstructions and the recovery rate of failed cases. We have added this analysis to Appendix A.3.1. ([***link***](https://anonymous.4open.science/w/hoi123touch-6121/#update_img23d). )
>
> ### 2. Comparisons with Grasp Generation Methods
>
> Grasp generation methods often rely on grasp-specific priors (e.g., pre-defined contacts like fingertips, palm orientation, and wrist rules) and focus primarily on palmar contact. To model the much broader and higher-dimensional space of free-form interactions, TOUCH differs in several ways.
>
> 1. TOUCH models the **entire hand surface (including the dorsal side)** and uses balanced resampling to enable diverse non-prehensile contacts.
> 2. Its **fine-grained control** is essential to prevent mode collapse to common grasping patterns, thereby ensuring interaction diversity.
> 3.  It **abandons grasp priors** and uses the cycle-consistency of contact for greater realism and flexibility.
>
> Conceptually, SemGrasp's multi-stage VQ-VAE architecture for grasp generation is similar to the ContactGen (a multi-stage VAE) baseline we already compare against. Furthermore, SemGrasp's contact guidance (e.g., "by four fingers") is general, lacking the explicit, fine-grained control over specific hand regions that is the core innovation of TOUCH.
>
> A direct comparison is infeasible due to SemGrasp's training code being unavailable and its reliance on a fine-tuned MLLM, making a fair comparison prohibitive. However, to address your valid concern for a more comprehensive evaluation, we have added new quantitative comparisons with two other relevant methods, NL2Contact and DiffH2O, further substantiating our claims.
>
> | metrics | P-IOU ↑ | P-F1 ↑ | MPVPE ↓ | PD ↓ | PV ↓ |  Ent. ↑ | CS ↑ | P-FID ↓ |
> | --- | --- | --- | --- | --- | --- | --- | --- | --- |
> | **Ours** | **0.776** | **0.844** | **2.97** | **0.932** | **2.67** |  **2.93** | **5.40** | **4.13** |
> | NL2Contact | 0.686 | 0.776 | 3.99 | 1.268 | 4.63 | 2.81 | 5.73 | 9.6 |
> | DiffH2O | 0.713 | 0.798 | 3.87 | 1.136 | 4.85 | 2.89 | 4.60 | 5.84 |
> -----
> Thank you again for your insightful comments. We hope our response has addressed your concerns.

---

### Meta-Review · Area_Chair_BqJH · 2026-01-06

**Summary:**

This paper proposes a text-guided framework for generating free-form hand–object interactions by curating large-scale in-the-wild videos and leveraging them to learn realistic grasp distributions and semantic hand–object behaviors. The core idea is conceptually simple, but the data curation process is non-trivial and carefully designed, and the resulting model demonstrates strong empirical performance across multiple tasks. Reviewers generally agreed that learning from in-the-wild videos provides meaningful benefits in capturing realistic interaction semantics and grasp diversity that are difficult to obtain from synthetic or lab-controlled data alone. Despite some concerns regarding evaluation completeness and claim strength, the overall direction and experimental results support acceptance.

**Reviewer Concerns:**

Concerns addressed or positively acknowledged:
Several reviewers highlighted the strength of the dataset curation effort, noting that mining in-the-wild videos enables the model to capture richer semantics and more realistic grasp distributions than prior work relying on synthetic or limited datasets. Reviewers also appreciated the breadth of experiments, including qualitative results that demonstrate diverse and semantically aligned hand–object interactions, as well as quantitative comparisons showing improvements over prior baselines. The integration of text guidance with hand–object generation was generally viewed as intuitive and effective, and the overall pipeline was considered well engineered.

Some reviewers also noted that, although the modeling components themselves are relatively standard, the paper’s value lies in the careful combination of data curation, representation design, and training strategy, which together lead to consistent performance gains. From this perspective, the work was seen as a solid empirical contribution rather than a fundamentally new modeling paradigm.

Existing concerns and suggested revisions:
A recurring concern across multiple reviews is the lack of a systematic evaluation of data quality. While the paper convincingly argues that in-the-wild videos provide better semantic coverage and more realistic grasp distributions, the analysis remains largely indirect. Reviewers suggested that more explicit measurements, such as comparisons of grasp diversity, noise levels, or annotation reliability, would strengthen the empirical claims. Addressing data quality in a more principled way is non-trivial, but doing so would significantly improve the paper.

Additional concerns include occasional over-strong claims relative to the simplicity of the core idea, and limited discussion of potential failure modes or biases introduced by in-the-wild data. These issues mainly affect framing and analysis depth rather than the correctness of the method, and can be addressed through clearer positioning and expanded discussion in the final version.

**Reviewer Scores:**

Reviewer JuZb: Likely to maintain their original score, as they were positive about the direction and empirical results, with their main concern being the lack of explicit data quality analysis.

Reviewer 8tRM: Likely to maintain their score after discussion, given the strong qualitative results and the convincing motivation for using in-the-wild video data.

Reviewer TWqj: Likely to maintain their score. While they noted that the core idea is simple, they acknowledged that the data curation effort is substantial and that the resulting performance gains justify acceptance.

Reviewer yAMZ: Likely to maintain their original score, viewing the work as a solid empirical contribution with clear practical relevance, while recommending additional analysis of data quality and limitations.

---

### Decision · Program_Chairs · 2026-01-26

Accept (Poster)